# The effect of the Pliocene temperature pattern on silicate weathering and Pliocene-Pleistocene cooling

Pierre Maffre[1], John C. H. Chiang[1], and Nicholas L. Swanson-Hysell[1]

[1]University of California, Berkeley

**Correspondence:** Pierre Maffre (maffre@berkeley.edu)

**Abstract.**

The warmer early Pliocene climate featured changes to global sea surface temperature (SST) patterns, namely a reduction to the equator-pole gradient and to the east-west SST gradient in the tropical Pacific, the so-called "permanent El Niño". Here we investigate the consequences of the SST changes to silicate weathering and thus to atmospheric $CO_2$ on geological timescales. Different SST patterns than today imply regional modifications of the hydrological cycle that directly affects continental silicate weathering in particular over tropical "hotspots" of weathering such as the Maritime continent, thus leading to a "weatherability pattern effect". We explore the impact of Pliocene-like SST changes on weathering using climate model and silicate weathering model simulations, and deduce $CO_2$ and temperature at carbon cycle equilibrium between solid Earth degassing and silicate weathering. In general, we find large regional increases and decreases to weathering fluxes and the net effect depends on the extent to which they cancel. Permanent El Niño conditions lead to a small amplification of warming relative to the present-day by $0.4\,°C$, suggesting that the demise of permanent El Niño could have had a small amplifying effect on cooling from the early Pliocene into the Pleistocene. For the reduced equator-pole gradient, the weathering increases and decreases largely cancel leading to no detectable difference in global temperature at carbon cycle equilibrium. A robust SST reconstruction of the Pliocene is needed for a quantitative evaluation of the weatherability pattern effect.

## 1 Introduction

Evidence has been accumulating that east-west (zonal) gradients of sea surface temperature (SST) and thermocline depth in the tropical Pacific were reduced during the warm early Pliocene ($4.5 - 3\,\mathrm{Ma}$) with respect to modern conditions (e.g., Cannariato and Ravelo, 1997; Ravelo, 2004; Wara et al., 2005; Fedorov et al., 2013). These features, resembling modern El Niño events, have led to the idea of a "permanent El Niño" climatic state during the Pliocene epoch, also named "El Padre" (Shukla et al., 2009). Paleoclimate proxy data also supports that the average equator-pole (meridional) temperature gradient was reduced during the warm early Pliocene (Brierley et al., 2009; Brierley and Fedorov, 2010; Fedorov et al., 2013).

The transition between early Pliocene to modern climate conditions is thought to be linked to global cooling from the Pliocene (5.3 to 2.6 Ma) into the Pleistocene (2.6 to 0.01 Ma), both as a cause or a consequence. This global cooling since the Pliocene has occurred while there is little evidence of significant change in $CO_2$ outgassing (e.g., Herbert et al., 2022), suggesting another trigger. Molnar and Cronin (2015) argued that the onset of a modern tropical Pacific zonal SST gradient (due to the

larger cooling of the eastern tropical Pacific) may have permitted the inception of the Laurentide ice sheet, promoting global cooling. In this hypothesis, teleconnection altered the atmospheric circulation over North America thereby enabling accumulation of perennial snow in winter in northern North America. The authors developed the hypothesis that vigorous modern Pacific Walker circulation was caused by the restriction of the Indonesian seaway as well as the increased land area of emerged islands in the Maritime Continent, both due to tectonic processes associated with arc-continent collision. The Walker circulation is an east-west atmospheric overturning circulation near the equator, that interacts with an oceanic counterpart, setting up a zonal SST gradient across the ocean basin. On the other hand, Fedorov et al. (2006) proposed a feedback loop where shoaling of the thermocline with cooling climate activates the Bjerknes feedback – the amplification of the cooling of the eastern boundary of a equatorial basin through upwelling of cold water due to easterly winds. This feedback promotes easterlies thus invigorating the Walker circulation and further enhancing climate cooling through cloud-albedo and ice-albedo feedback. The zonal tropical Pacific SST gradient has also been shown to be linked to the global meridional temperature gradient (Fedorov et al., 2015). Therefore, all the following elements, strengthening zonal and meridional gradients, intensification of El Niño Southern Oscillation (ENSO), onset of a mean climate state closer to La Niña, and global cooling, would be connected processes.

One element that has been largely overlooked in these considerations about regional climate changes is their effect on silicate weathering and the silicate weathering feedback. The so-called silicate weathering paleothermostat (Walker et al., 1981) is the hypothesis that long-term climate is controlled by the balance between deep earth degassing of $CO_2$ and its consumption by continental silicate weathering and associated oceanic carbonate precipitation (the Urey reaction, Urey, 1952). Weathering reactions are enhanced by warmer climate which introduces a negative feedback: atmospheric $CO_2$ is stabilized at the level where its consumption balances its degassing (Walker et al., 1981; Berner et al., 1983). Continental runoff rate is a key factor controlling silicate weathering rate (Gaillardet et al., 1999; Dessert et al., 2003; Oliva et al., 2003; Maher and Chamberlain, 2014). The very existence of the silicate weathering paleothermostat relies to a large extent on the fact that runoff rates, on average, increase with rising $CO_2$ and associated global warming (Kukla et al., 2022). Hence, processes that affect the distribution of continental runoff, even without direct effects on global temperature, have the potential to change the efficiency of silicate weathering, and therefore, equilibrium atmospheric $CO_2$ level, and global temperature. This phenomenon is analogous to the so-called "pattern effect" (Langenbrunner, 2020) in the Earth's temperature climate sensitivity to atmospheric $CO_2$. As with climate sensitivity, SST patterns may have significant effects on the efficiency of silicate weathering – hereafter called "weatherability" – because they affect atmospheric convection, precipitation and runoff.

In that regard, reducing meridional or zonal SST gradients (with respect to modern climate) may decrease global weatherability by shifting rainfall away from tropical land masses. In the case of meridional gradient, Burls and Fedorov (2017) suggested a reduced equatorward moisture transport by the Hadley circulation with flatter SST gradient. This feature can be summarized by "wet gets dryer, dry gets wetter" (Burls and Fedorov, 2017). In the case of zonal gradients, a weaker Pacific gradient should lead to reduced Pacific Walker circulation, that is responsible of intense precipitation over the Maritime continent in modern climate. Indeed, El Niño events, where the Pacific Walker circulation collapses, leads to a drastic decrease of continental runoff on the Maritime continent, as well as most of South America near the equator (see Fig. 1 showing the El Niño-years climatology of precipitation and runoff). On average, runoff is reduced by $\sim 3\%$ on land during El Niño-years. The key element here is

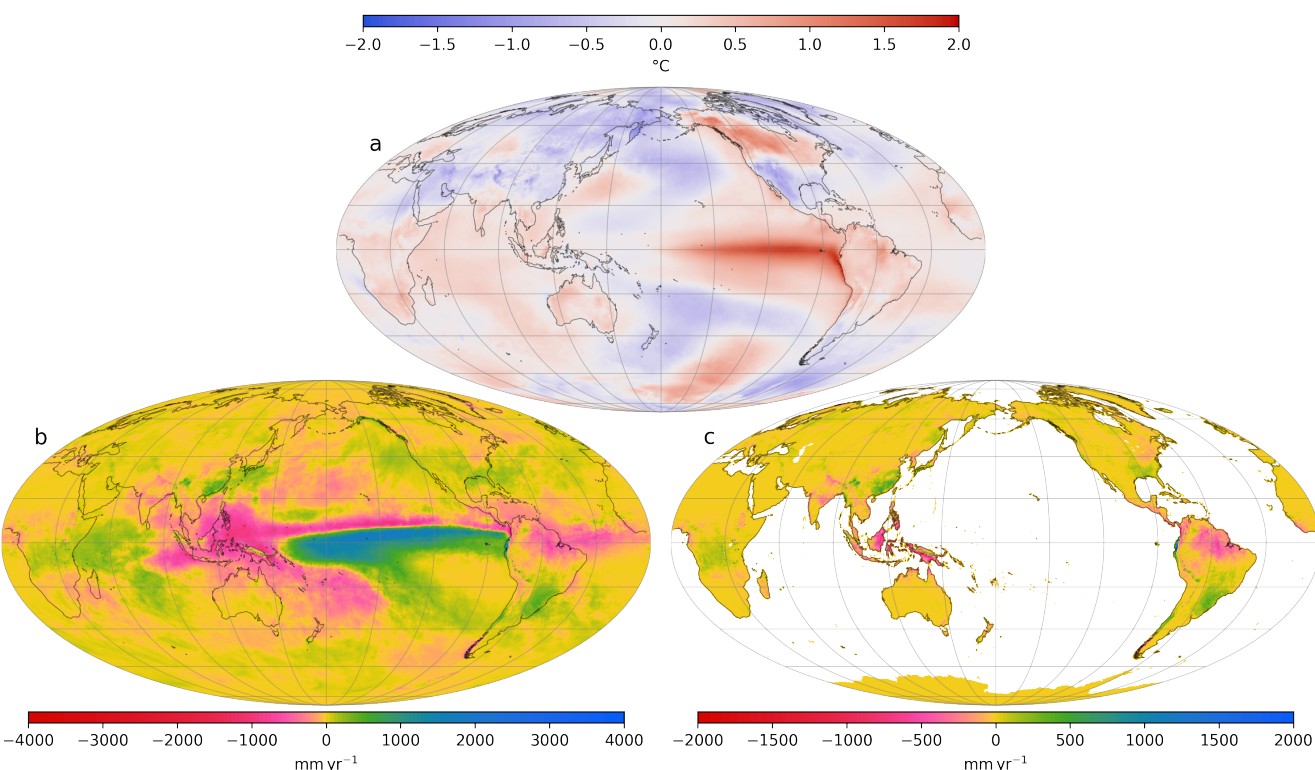

**Figure 1.** Differences in climatology associated with El Niño years: average of selected El Niño years minus average of years selected neither as El Niño nor as La Niña. Each "year" considered here is from May to the following April. Data is from ERA5 reanalysis. See Methods, section 2.2, and Appendix A for details. (a): 2 m temperature, (b): total precipitation, and (c): continental runoff. On each panel, non-significant differences are hatched (p-value of Welch's t-test $> 0.1$, see Appendix C). The maps are Mollweide projections with graticules plotted every $30°$ of longitude and every $20°$ of latitude (both starting from $0°$). Note that it is significantly drier over the South-East Asian Islands in El Niño years with more precipitation falling over the tropical Pacific ocean rather than over land.

that some tropical areas that are drier in El Niño years, are known hotspots of silicate weathering – mainly, the Southeast Asian Islands known as the Maritime continent (Figs. 1 and 2a). Another contribution to changes in silicate weathering should be expected from the Himalaya and the portion of the Andes north of the equator, that also experience decreases in runoff during El Niño years.

The above text and associated hypothesis can be summarized as: "With weaker meridional and zonal temperature gradients, the decrease of runoff over land, specifically on tropical weathering hotspots, would lead to reduced weatherability and, as a result, higher $CO_2$ levels contributing to the warmer Pliocene climate." If we further assume that colder climate favors stronger meridional and zonal gradients, the overall process of cooling would act as a positive feedback loop where increased cooling leads to increased precipitation on key land masses, which would lead to enhanced weatherability and increased cooling. This

process has the potential to have amplified climate cooling from the warm early Pliocene into the Pleistocene.

In this contribution, we investigate the potential effect of meridional and zonal gradients on global climate through the silicate weathering feedback, without seeking to determine the cause of these regional climate features. We first provide an estimation of the silicate weathering anomaly using the climate fields from ERA5 reanalysis 1979–2020 (Muñoz Sabater, J., 2019; Hersbach et al., 2019), with the assumption that the average climate of El Niño events can be used as a proxy for Pliocene permanent El Niño. This method enables a qualitative assessment of whether the effect of a permanent El Niño state on global chemical weathering would generate a warmer or colder global climate (because of a decrease or an increase of chemical weathering, respectively). However, one cannot quantitatively estimate this warming or cooling because the climate reanalysis only exist at current $CO_2$ level. Yet, any disturbance in the efficiency of silicate weathering – that is, increase or decrease of silicate weathering flux at current $CO_2$ level – will change $CO_2$ until the balance between $CO_2$ degassing and its consumption by silicate weathering is restored. On the million-years timescale of the Pliocene to the modern, it is this equilibrium $CO_2$ level that matters for climate evolution. Furthermore, it is possible that the average of El Niño years is not a perfect analog of a permanent El Niño, because of the transient nature of those events, and the fact that they are not bound to radiative balance, as is a long-term equilibrium climate state.

For these two reasons, we designed numerical climate experiments aimed at exploring Pliocene climate features with a climate model, focusing on the meridional and the zonal tropical SST gradient. Doing so, we provide an initial quantitative estimation of the role of a "weatherability pattern effect" on climate evolution since the warm early Pliocene, through the silicate weathering feedback.

## 2 Methods

### 2.1 Silicate weathering model

We use the silicate weathering model published in Park et al. (2020). This model represents the steady-state vertical profile of primary silicate minerals abundance in a regolith – the interface between unweathered bedrock and land surface. Minerals are progressively weathered until they are removed from the regolith through physical erosion of the surface. Hence, weathering profiles show a continuous depletion of primary minerals from bedrock to the surface. The vertically-integrated weathering rate depends on the rate of physical erosion (which leads to upward advection of primary minerals), the efficiency of weathering chemical reactions (that depend on runoff and temperature), and the residence time of minerals in the regolith (that is also controlled by the physical erosion rate). The derivation of this model was carried out by Gabet and Mudd (2009) and West (2012), who parameterized the climatic control of weathering reactions (through temperature and runoff). The erosion rate is computed as a power-law of topographic slope and runoff rate (Maffre et al., 2018; Park et al., 2020).

The value of interest computed by the weathering model is the amount of dissolved $Ca$ and $Mg$ released by weathering of silicate minerals (integrated over the regolith profile). This flux corresponds to long-term $CO_2$ consumption in the geological carbon cycle through the eventual precipitation of marine carbonates. The weathering model is applied on every point of the continental mesh grid, at the resolution of the climate fields used (see following sections). On each point, computation of the estimated weathering rate is done for 5 lithological classes of silicates (as in Park et al., 2020) that are simplified from the global

lithological map of Hartmann and Moosdorf (2012). The slope field was derived from the elevation dataset of the Shuttle Radar Topography Mission (Farr et al., 2007) at 30" resolution. Both the lithological classes and the slope were interpolated on the desired mesh grids.

The model parameters are taken from Park et al. (2020) who provided an ensemble of 573 "best-fit" unique parameter combinations resulting from a comparison of model results to modern chemical weathering fluxes across watersheds. As in Park et al. (2020), the weathering model is run with all those unique parameter combinations. We present, for each model run, the ensemble of results (e.g., the ensemble of the global weathering flux). This approach allows us to quantify the uncertainties arising from the weathering parameters.

The silicate weathering model does not explicitly represent the role of vegetation as a weathering agent. Yet, because it is calibrated with data from natural systems, the parameters incorporate a part of the vegetation effects embedded in the data. For instance, the values of parameters determining the sensitivity to the runoff ($k_d$ and $k_w$) are influenced by the correlation between water availability and the presence of vegetation.

## 2.2 Climate reanalysis and selection of El Niño and La Niña years

The reanalysis climate fields used in this study are from ERA5-land 1981–2019 (Muñoz Sabater, J., 2019) for continental runoff, and ERA5 1979–2020 (Hersbach et al., 2019) for $2\,\mathrm{m}$ temperature and SST. The monthly averaged fields at the native resolution ($0.1\,^\circ$ for ERA5-land and $0.25\,^\circ$ for ERA5) were interpolated on a longitude-latitude grid of $0.5\,^\circ$ resolution.

To determine the climatology associated with El Niño and La Niña years, we generated an interannual index of ENSO, and then used it to create selections of El Niño years, La Niña years, and neither El Niño nor La Niña years. The details of the calculation of this ENSO index can be found in Appendix A. Based on this index, we selected the years 1982, 1987, 1991, 1997, 2009 and 2015 as El Niño years, and the years 1984, 1988, 1999, 2007 and 2010 La Niña years (in each case, from May to April of the next calendar year).

## 2.3 Climate model

For the climate numerical experiments, we used the Community Earth System Model (CESM) version 1.2.2.1. The experiments were conducted using the components CAM4 for the atmospheric dynamics, CLM4.0 land model, CLM4.0 CN river-runoff component, CICE prognostic sea ice, and for the oceanic component, either fixed SSTk DOCN in slab ocean mode, or full ocean model (Parallel Ocean Program version 2). The slab ocean approximates a well-mixed ocean mixed layer with a fixed depth set to the annual mean; an ocean heat transport convergence (i.e., the "Q flux") from a CESM1 pre-industrial fully coupled run is prescribed at each ocean gridpoint to achieve a simulated sea surface temperature close to the pre-industrial. The grids used are a regular $0.9\,^\circ \times 1.25\,^\circ$ (latitude$\times$ longitude) for atmosphere and land modules, a regular $0.5\,^\circ$ for runoff routing, and a $\sim 1\,^\circ$ "displaced" grid, with Greenland pole, (0.9x1.25_gx1v6) for the ocean models (both slab and full) and sea ice modules.

All experiments were conducted with permanent 1850 (pre-industrial) boundary conditions, with the exception of the few modifications discussed later in the article, that are atmospheric $CO_2$ concentration, slab ocean Q flux, and clouds albedo.

Experiments were run 40 years for the fixed SST cases, 50 years for the slab ocean cases, and 230 years (170 years for the control) for the coupled ocean-atmosphere cases. All experiments were initiated with a "cold start" (internally-generated initial condition, corresponding to a pre-industrial climate). In all cases, the last 30 years of the model runs were used to compute the climatology average. See Table B1 and Appendix B for more information about the specific features of the climate simulations conducted here.

## 2.4 Geographic division of silicate weathering

To quantify the contributions of the different regions of Earth to the changes of silicate weathering flux due to perturbations of climate fields, we used the geographic division shown in Fig. 2a. This division is meant to isolate known weathering hotspots (e.g. Maritime Continent and East African rift), major mountain ranges, and, for the remaining regions, tropics versus subtropics. The boundaries of these regions are somewhat arbitrary, especially for the mountain ranges, and their naming should be considered as a simplification of the broader regions. For instance, "Himalaya" refer to the merging of the Himalayan range, its extension over the Longmen Shan and in South-East Asia, the Tibetan plateau and the Tian Shan. Despite the subjectivity of those boundaries, these divisions give a sense of the different response of individual regions that are useful for summarizing results.

## 3 Results

### 3.1 Weathering rates using ERA5 reanalysis

#### 3.1.1 Control simulation

We first run the weathering model using ERA5 reanalysis of climate fields (1981–2019 climatology average), as a control simulation. The resulting weathering rates (averaged over the selected parameterizations) are presented in on a map in Fig. 2b. The total $CO_2$ consumption by weathering is $5.3\,\mathrm{Tmol\,yr^{-1}}$. This control weathering field is very similar to the one published in (Park et al., 2020, Fig. S5 of their contribution), who used the same model, parameter combinations and slope fields, but different climate fields (CRU TS v.4.03 for the temperature, Harris et al. (2014), and UNH/GRDC Composite Runoff Fields V1.0, Fekete et al. (1999)).

The contribution of the different regions of Earth to this control weathering flux is shown in Fig. 2c (blue bars). The weathering flux from the Maritime Continent is about 11% of the total flux, which is consistent with previous estimates (e.g., Molnar and Cronin, 2015). The main mountain belts (broader Himalaya, Andes and North and central American ranges, including the rocky mountains) together contribute to $\sim 22\%$ of the total. Another significant weathering contributor is the East African Rift ($\sim 8\%$ of the total), owing to relatively high runoff rates, the substantial amount mafic silicate rocks exposed, and the relatively steep topography that fosters physical erosion – and hence, elevated chemical weathering fluxes. Taiwan and New Zealand, that are known hotspots of physical erosion rates, do not contribute significantly to the total chemical weathering flux because of the limited areal extent of their orogens.

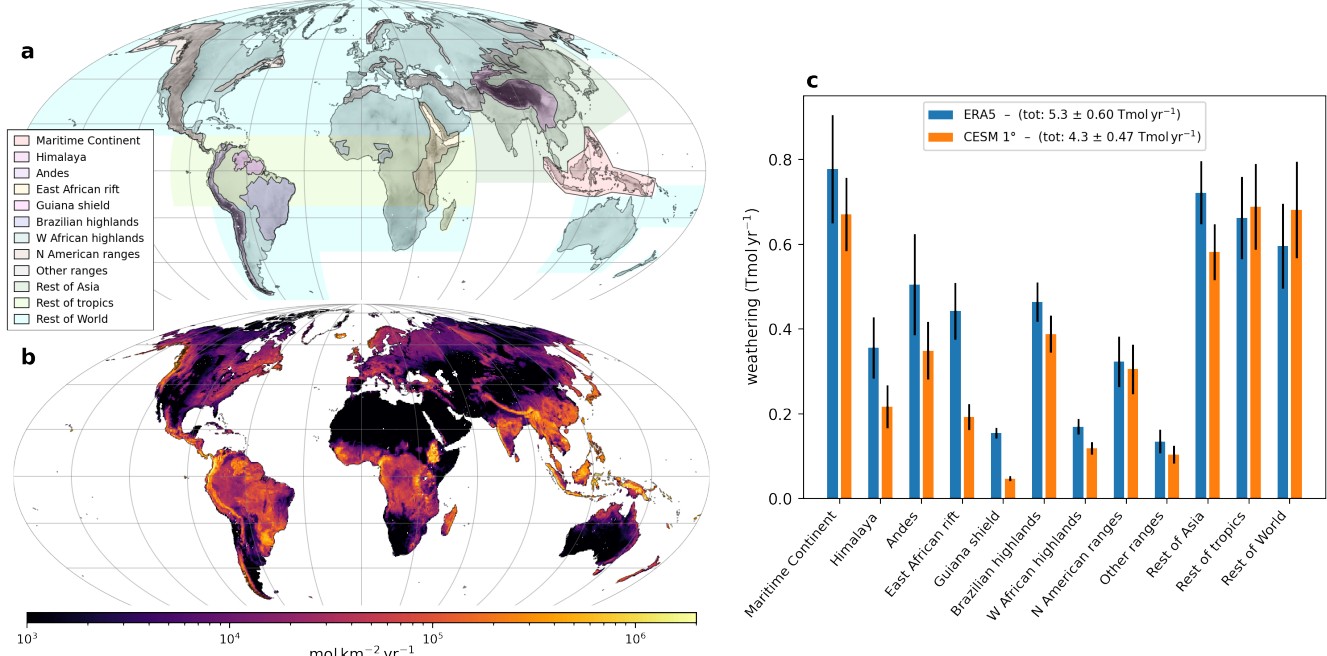

**Figure 2.** (a): Map of the geographic division used to summarize results in this study (colored semi-transparent shades) superimposed on land topography. (b): Average chemical weathering rates ($Ca$ and $Mg$ from silicate minerals) over the parameter combinations (i.e., the weathering model is run for each of the 573 selected parameter combination, and the 573 weathering fields are then averaged). Weathering is computed using climate fields from ERA5 reanalysis (climatology average 1981–2019) at $0.5\,^\circ$ of resolution. (c): Bar plot of chemical weathering fluxes integrated over the geographic regions (panel a), using ERA5 climate fields – as for panel (b) (blue bars); or using climate fields from CESM1.2 climate model pre-industrial control simulation, with slab ocean model, performed in this study (average of last 30 years simulations; orange bars). The main colored bars denote the average weathering flux of the 573 parameterizations, whereas the black error bars denote the standard deviation over those parameterizations. The total (summed) weathering flux is indicated on the legend box giving both the average and associated standard-deviation resulting from the different weathering model parameterizations. The maps projection and graticules (panels a and b) are similar to Fig. 1

### 3.1.2 El Niño and La Niña years climatologies

We then recompute the weathering rates with the same dataset, but keeping only the years of the climate time-series identified
as "El Niño" or "La Niña" (average of May to April, see Methods, section 2.2). The anomaly of Earth-integrated weathering
flux is $-90 \pm 15.4\,\mathrm{Gmol\,yr^{-1}}$ for El Niño years and $+138 \pm 30.1\,\mathrm{Gmol\,yr^{-1}}$ for La Niña ("$\pm$" denoting the standard deviation
of the weathering anomaly over the selected parameterizations). These calculations are assuming that these climatologies are
imposed over a long-timescale such that the chemical weathering profiles adjust to result in these fluxes. Assuming that $CO_2$
degassing is constant, these weathering anomalies would in reality be compensated by global warming or cooling, so that the
net silicate weathering flux anomaly is zero. If we extrapolate from the silicate weathering feedback computed with CESM

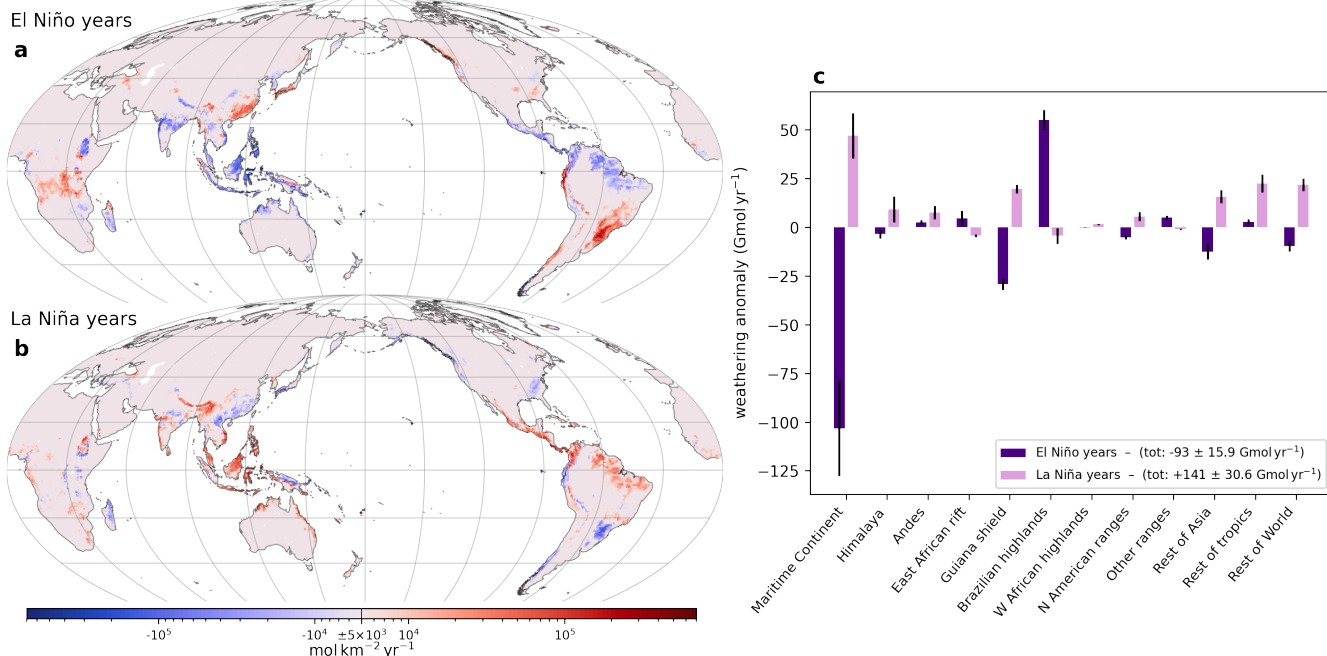

**Figure 3.** (a) and (b): Anomaly of modeled weathering rate with respect to control weathering (i.e., the field shown in Fig. 2b) for El Niño years climatology (a) and La Niña years climatology (b). The weathering rate is averaged over the parameter combinations (as in Fig. 2b). The colorbar scale is a "symmetric logarithm", and any value lower than $5 \cdot 10^3 \,\mathrm{mol\,km^{-2}\,yr^{-1}}$ (in absolute value) is approximated as 0. Map projection and graticules are as in to Fig. 1. (c): Bar plot of the anomaly of weathering flux integrated over the geographic regions shown in Fig. 2a with respect to ERA5 control weathering (blue bars in Fig. 2c) for El Niño or La Niña years climatology. Average (main colored bars) and standard deviation (error bars) refer to the parameterizations, as in Fig. 2c. There are many regional changes of note as discussed in the text. For example, there is enhanced weathering over the Maritime continent during La Niña years relative to El Niño years. The change in weathering rate is most dramatic over the Maritime Continent due to increased precipitation over land during La Niña years compared to being over the Pacific Ocean during El Niño years.

climate simulations, indicating a global weathering flux increase of $\sim 0.4 \,\mathrm{Tmol\,yr^{-1}}$ per $^\circ$C of global warming by $CO_2$ (Fig. 8b, solid black curve), the weathering anomalies should then be compensated by global mean temperature anomalies of $+0.23\,^\circ$C for long-term El Niño conditions and $-0.35\,^\circ$C for long-term La Niña conditions. These relatively small values of weathering anomaly (1.7% and 2.6% of the total control flux, respectively) are the consequences of local increase and decrease of weathering rates that compensate each other (Fig. 3). Nevertheless, the differences from 0 are robust among all the selected parameterizations (the absolute anomaly of weathering flux is $\sim 4$ times larger than the standard deviation due to the parameter uncertainties), and approximately symmetrical between El Niño and La Niña conditions. These results support our hypothesis of climate cooling caused by the progressive onset of "La Niña-like" mean climatic conditions, though the magnitude of this cooling may not have been substantial.

Local weathering anomalies are largely driven by changes in runoff rates. Many regions display dipolar patterns under El Niño conditions (Fig. 3a). For example, weathering fluxes increase on the Ethiopian Traps associated with El Niño conditions, but decrease on the southern part of the East African rift, they increase on the Andes near the equator, but decrease in the northernmost part of the range, and decrease on the central Himalaya but increase the western part. Hence, the anomaly of weathering flux integrated over those regions is close to zero (Fig. 3c). Most of the negative anomaly of El Niño weathering

comes from the Maritime Continent. The weathering integrated over this region is almost equal to the total anomaly (Fig. 3c, purple bars). This result highlights the key role this area has potentially played in cooling Earth's climate over the last few million years. The signal from the Maritime Continent is expanded by a weathering decrease on the Deccan Traps and the Guiana highlands associated with El Niño conditions, and partially offset by an increase on the Brazilian highlands (Fig. 3).

    It is interesting to note that although the sign of weathering anomalies under La Niña conditions is nearly everywhere the

195 opposite of the ones under El Niño conditions, the magnitude of those anomalies is not perfectly symmetrical (Fig. 3c).

### 3.2   Exploring the consequences of Pliocene SST

We wish here to take a step further and investigate the consequence of Pliocene-like gradients of SST on silicate weathering and equilibrium temperature (i.e., the global mean temperature for which silicate weathering balances degassing). We rely here on the work of Fedorov et al. (2015) demonstrating that the tropical zonal gradient of SST is linked to the meridional gradient

of SST. This result indicates that Pliocene permanent El Niño could be a consequence of weakened meridional gradient. Drawing from their work, we designed climate simulations aimed at mimicking Pliocene SST, following the method of Burls and Fedorov (2014).

    The climate simulations presented in Burls and Fedorov (2014), Fedorov et al. (2015), and Burls and Fedorov (2017) cannot be directly used because the simulations at $0.9\,^{\circ} \times 1.25\,^{\circ}$ of resolution were only conducted at pre-industrial $CO_2$. However,

estimating carbon cycle equilibrium with GEOCLIM necessitates climate simulations at several $CO_2$ levels. This cannot be done with fixed SST simulations (such as from Fedorov et al., 2015), because it will miss the warming of ocean surface with rising $CO_2$. On the other hand, changing atmospheric $CO_2$ in a coupled ocean-atmosphere model at $0.9\,^{\circ} \times 1.25\,^{\circ}$ would require a prohibiting computation time to let the full ocean be at equilibrium with the radiative forcing, and get the correct atmospheric warming. Another issue is that Burls and Fedorov (2014) produced their Pliocene-like SST pattern by altering the

cloud properties in the cloud parameterization; as such, the physics of that simulation differs from that of the the pre-industrial control, and thus the two simulations are not directly comparable.

    Because of these constraints, we designed slab ocean climate simulations that reproduce the Pliocene-like SST pattern of Burls and Fedorov (2014), simulations that can be run at many $CO_2$ levels. We first followed the method of Burls and Fedorov (2014) of altering the cloud radiative properties with a coupled ocean-atmosphere model. From those simulations, we generated

a Q flux field that we used to force the slab ocean version of the climate model and reproduce the SST pattern of the coupled ocean-atmosphere simulations. The generation of Q flux field implied conducting intermediate atmosphere-only simulations, this explained in details in Appendix B.

Two Q flux fields were actually generated, one for reproducing the entire SST pattern ("full Pliocene SST") and one for reproducing the SST pattern only between $10\,°$S and $10\,°$N ("10 °SN Pliocene SST"). The motivation for isolating the SST pattern close to the equator is that El Niño events consist of the collapse of the tropical Pacific zonal gradient of SST, without major changes in extra-tropical SST patterns – though extra-tropical teleconnections do modify local SST – while Pliocene climate proxies indicate a reduction of both zonal and meridional gradients. Yet, El Niño is viewed as analogy for Pliocene climate, in particular, the same teleconnections are assumed to occur (Molnar and Cane, 2002). Also, this configuration allows us to compare the effects of a permanent El Nino to that of the long-term average of transient El Ninos in section 3.1.2. Isolating the $10\,°$S–$10\,°$N SST pattern allows us to investigate the teleconnections caused to the tropical part of the Pliocene SST estimate, without the effects of the reduced meridional gradient, to better compare it to El Niño events.

Figure 4 shows the comparison of the SST anomaly (with respect to pre-industrial control) taken from the coupled ocean-atmosphere simulations and the SST anomaly from the slab ocean simulations, for both the "full" and "10 °SN" cases. SST proxies from 10 ODP sites (see Appendix D) are also shown to Fig. 4 a and b. The comparison to the proxies confirm the results of Fedorov et al. (2015) that the alteration of clouds radiative properties reproduce the mid- and high-latitude warming, and the Eastern tropical Pacific warming. This last warming, however, is not as pronounced ($\sim 2.5\,°$C) as indicated by the proxies ($\sim 4\,°$C. See Fig. 4b).

The slab ocean simulations reproduce well the SST patterns from the coupled ocean-atmosphere simulations, except in a few local spots. For instance, on Fig. 4 b and d, it appears that the "10 °SN Pliocene SST" slab ocean simulation fails to generate a strictly null SST anomaly outside the $10\,°$S–$10\,°$N band, but the magnitude of those subtropical anomalies is negligible compared to the expected ones within $10\,°$S–$10\,°$N. Regions influenced by sea-ice exhibit significant local differences, but they do not alter the global behaviour of the simulations.

### 3.2.1 Weathering and equilibrium temperature with reduced zonal gradient

In this section, we examine the simulations where the Pliocene SST field is applied in the $10\,°$S–$10\,°$N band (10 °SN Pliocene SST) with the slab ocean version of CESM.

The standard $CO_2$ experiment for this configuration ($299.4\,\mathrm{ppmv}$, see Appendix B, section B2) has a Global Mean $2\,\mathrm{m}$ Temperature of $13.67\,°$C, which is $0.25\,°$C warmer than the pre-industrial control (at $284.7\,\mathrm{ppmv}$). The precipitation anomaly (Fig. 5a) exhibits a large increase in the eastern Pacific, similar to El Niño events. This increase does not extend southward as much as for observed El Niño conditions (Fig. 1b), likely because we imposed SST perturbation only in the $10\,°$S–$10\,°$N band. Instead of drier conditions around the Maritime Continent and wetter conditions in the western Indian ocean characteristic of El Niño, precipitation increases along the equator ($\sim \pm 5\,°$) across the Indian ocean, and extends up to the eastern coast of Borneo over the Maritime Continent. The islands of Borneo, Sulawesi, and New Guinea show a similar pattern of wetter conditions except on their western edge. The tropical Atlantic also shows significant difference with El Niño, as because the imposed SST in the Atlantic is warmer almost uniformly along the equator (Fig. 4 b and d), which favors atmospheric convection. The runoff anomaly (Fig. 5 a) reveals some similarities with the El Niño case (Fig. 1c): a decrease over India and South-East Asia, an increase over China and the eastern side of the Himalaya, an increase in the Andes around the equator, a decrease in the rest

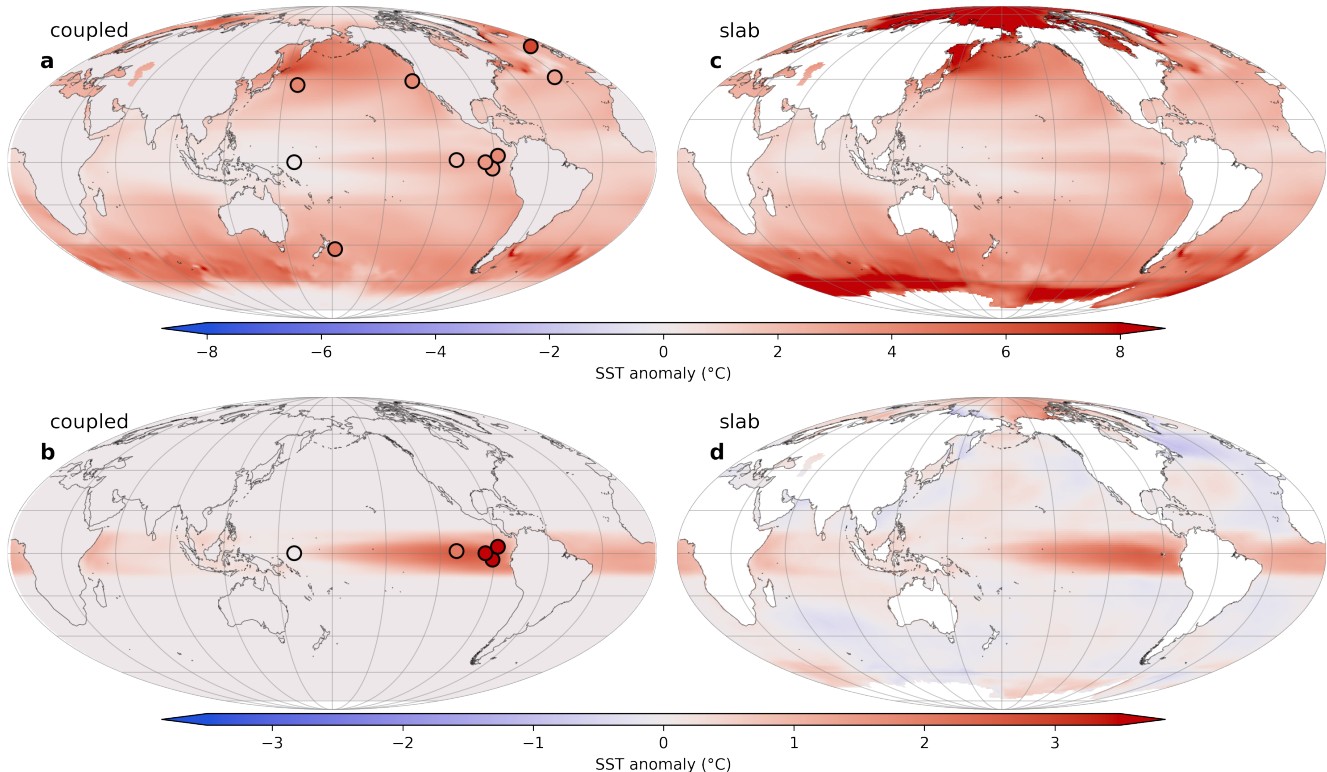

**Figure 4.** Anomaly of SST (a and b) and ocean skin temperature – i.e., SST, except on grid points with sea ice (c and d)). (a): coupled ocean-atmosphere simulation with altered cloud properties (i.e. full Pliocene SST), pre-industrial $CO_2$. (b): same as (a) but SST anomaly field is truncated at $10\,°$ N or S (i.e., $10\,°$SN Pliocene SST). (c): slab ocean simulation with Q-flux derived to reproduce the full Pliocene SST (i.e., SST of panel a; $CO_2$ at 300 ppmv). (d): slab ocean simulation with Q-flux derived to reproduce the $10\,°$SN Pliocene SST (i.e., SST of panel b; $CO_2$ at 299.4 ppmv). Note that the scale in (a) and (c) is different from (b) and (d). Colored dots in (a) and (b) are estimates of SST anomaly at $4.5$–$3.5\,\mathrm{Ma}$ from ODP proxies (see Appendix D). They follow the same colorscale than their underlying map. Map projection and graticules are similar to Fig. 1. Large temperature anomalies ($> 8\,°\mathrm{C}$) in (c) are found in regions influenced by sea ice, and deviate from the original SST (panel a), that does not respond to the cloud albedo forcing (SST anomaly is $0\,°\mathrm{C}$) because of the sea ice cover. Note that in (d), the temperature anomaly is not perfectly $0\,°\mathrm{C}$ outside $10\,°$S–$10\,°$N, because of inaccuracies in the derivation of the Q flux, but the amplitude of those anomalies is significantly less than within $10\,°$S–$10\,°$N, with the exception of sea-ice regions.

of equatorial South America (though the enhanced convection in the Atlantic generates wetter conditions along the Atlantic coast), an increase in the South-east part of South America, and an increase over the Congo basin. The largest discrepancies are found on the Maritime Continent, and the East African rift.

We then used the $10\,°$SN Pliocene SST simulations at several $CO_2$ levels to compute the equilibrium $CO_2$, which is the value at which the silicate weathering flux with $10\,°$SN Pliocene SST balances $CO_2$ degassing. More precisely, we consider that control $CO_2$ degassing is equal to pre-industrial silicate weathering flux. Therefore, for each selected combination of

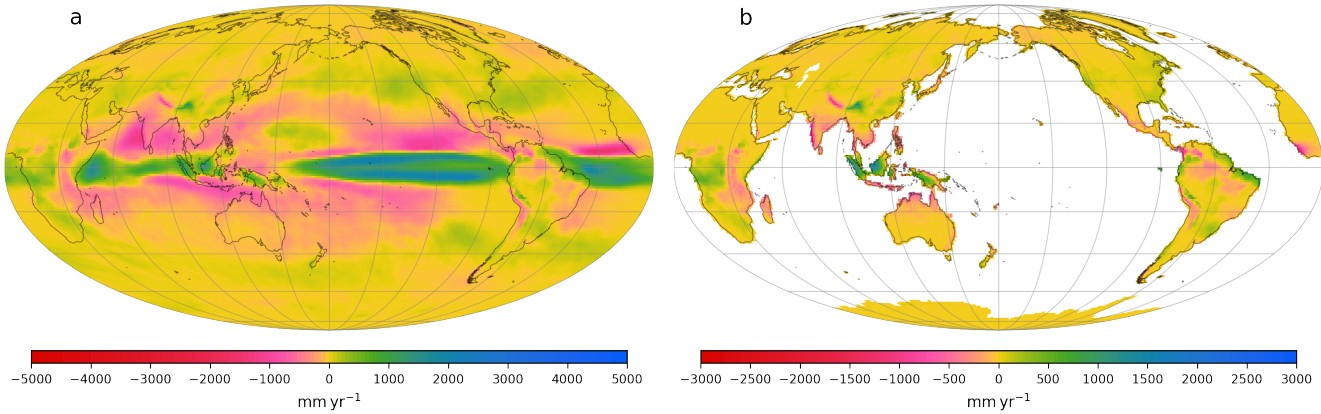

**Figure 5.** Annual mean climatology of slab ocean simulation with $10\,°\mathrm{SN}$ Pliocene SST at $299.4\,\mathrm{ppmv}$ anomaly with respect to the pre-industrial control. (a): total precipitation, and (b): continental runoff. Map projection and graticules as in Fig. 1. On each panel, non-significant differences are hatched (p-value of Welch's t-test $> 0.1$, see Appendix C)

GEOCLIM parameters (see Methods, section 2.1), we imposed that the silicate weathering flux with $10\,°\mathrm{SN}$ Pliocene SST is equal to the silicate weathering flux with the same parameter combination in the pre-industrial control simulation (i.e., computed with climate fields from pre-industrial slab simulation. This "control" weathering is shown on Fig. 2c, orange bars.) Figure 6 shows this equilibrium $CO_2$ (panel a) and the Global Mean $2\,\mathrm{m}$ Temperature corresponding to that $CO_2$ concentration (panel b). The Pliocene SST (within $10\,°\mathrm{S}$–$10\,°\mathrm{N}$) generates climatic conditions less favourable for weathering, causing $CO_2$ to increase, and global temperature to rise by $\sim 0.4\,°\mathrm{C}$, which is about twice that estimated using ERA5 reanalysis of El Niño years. This result is consistent with the fact that, in the $10\,°\mathrm{SN}$ Pliocene SST simulation, the west-to-east Pacific SST difference is reduced by more ($\sim 2.5\,°$) than in the El Niño years climatology ($\sim 1.5\,°$). Note that, though the climate sensitivity (global temperature difference for a $CO_2$ doubling) stays the same, there is an offset in the $CO_2$-temperature relationship, meaning that a $CO_2$ of $284.7\,\mathrm{ppmv}$ does not correspond to a $0\,°\mathrm{C}$ temperature anomaly (see also Fig. 8a).

This decrease of weatherability (or weathering efficiency) can be investigated by looking at the anomaly of silicate weathering rate of $10\,°\mathrm{SN}$ Pliocene SST at $299.4\,\mathrm{ppmv}$ with respect to pre-industrial control weathering rate (Fig. 7). Though the global temperature is not the same ($10\,°\mathrm{SN}$ Pliocene SST is $0.25\,°\mathrm{C}$ warmer), it still illustrates the causes of this decrease in weatherability. Similar to what was observed with El Niño and La Niña composites (Fig. 3), several positive and negative regional anomalies of weathering, driven by runoff changes, compensate each other, leading to a weak warming. Nevertheless, this warming is consistent over the whole ensemble of parameter combinations. Figure 7b show the integration of the weathering rate anomaly from Fig. 7a over the geographic regions presented Fig. 2a. Unlike the El Niño case, the weathering anomaly integrated over the Maritime Continent is positive, and thereby not contributing to the warming. Indeed, there are more increases than decreases in runoff on the Maritime Continent in the $10\,°\mathrm{SN}$ Pliocene SST simulation. The East African rift also has an opposite behaviour, with a decrease of weathering instead of a slight increase with El Niño. The last main difference

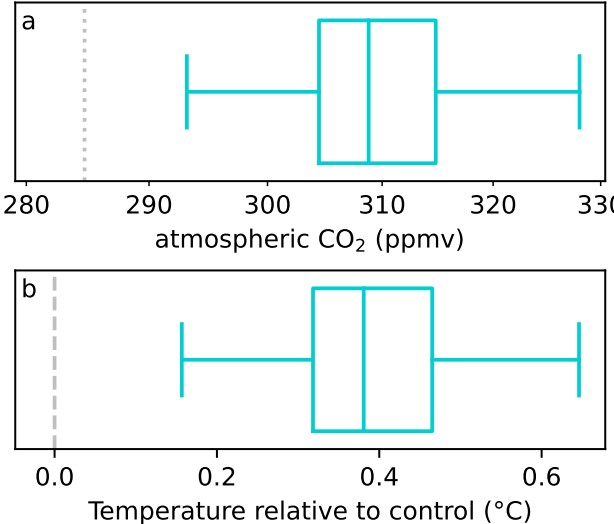

**Figure 6.** Results for the $10\,°\mathrm{SN}$ Pliocene SST GEOCLIM simulation at C cycle equilibrium (i.e., with silicate weathering flux balancing control degassing). (a): atmospheric $CO_2$ level (logarithmic axis) and (b): anomaly (with respect to pre-industrial simulation) of Global Mean $2\,\mathrm{m}$ Temperature. The distribution – shown as a boxplot – represents the ensemble of GEOCLIM parameterizations (similarly to Fig. 2c and 3c). The x axis of panel (a) is bounded so that a given $CO_2$ in panel (a) correspond to the aligned global temperature in panel (b). This with the approximation that the $CO_2$ is continuously logarithmic (from lower to upper bound), whereas GEOCLIM $\log(CO_2)$-interpolates climate fields between the $CO_2$ levels – the ones considered here being $250\,\mathrm{ppmv}$, $299.4\,\mathrm{ppmv}$ and $427.1\,\mathrm{ppmv}$. Because of this approximation, the boxplots in (a) and (b) are not perfectly aligned, but this misalignment is negligible. The pre-industrial $CO_2$ level and the $0\,°\mathrm{C}$ anomaly are highlighted by the thin grey dashed or dotted vertical lines. Note that those two lines do not align because of the offset in the $CO_2$–temperature relationship of the $10\,°\mathrm{SN}$ Pliocene SST simulation with respect to the pre-industrial one.

concerns the category "rest of Asia", showing a large decrease in weathering. The decrease in runoff in India, and associated weathering decrease, overwhelms the increase in China, whereas those two balance out under modern El Niño conditions.

The weatherability decrease of the $10\,°\mathrm{SN}$ Pliocene SST simulation can also be appreciated by looking at the global temperature–total weathering flux relationship (Fig. 8b). The $10\,°\mathrm{SN}$ Pliocene SST curve (dashed blue) is systemically beneath the pre-industrial one (solid black), for any given global temperature. This means that regardless of the chosen background $CO_2$ degassing flux, the simulation with $10\,°\mathrm{SN}$ Pliocene SST will be warmer.

In summary, applying our estimation of Pliocene SST in the $10\,°\mathrm{S}$–$10\,°\mathrm{N}$ band – whose main feature is a flattening the
tropical Pacific zonal gradient – generates a similar weatherability decrease and global warming to that estimated from El Niño (section 3.1.2), though approximately twice as large. However, the mechanisms of that weatherability decrease are different, due to differences in the regional patterns of simulated runoff.

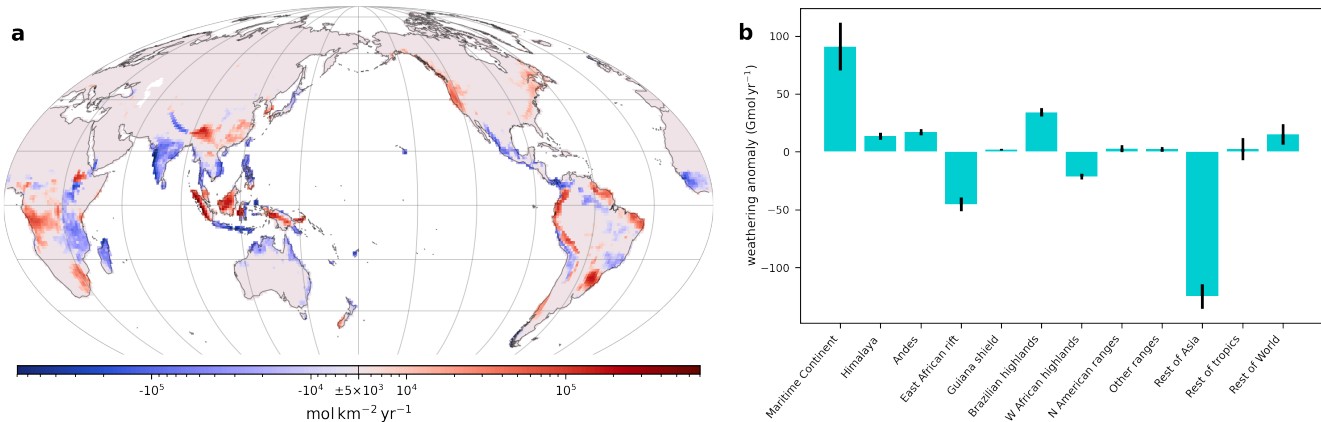

**Figure 7.** (a): Anomaly of weathering rate of slab ocean climate simulation with $10\,°$SN Pliocene SST at $299.4\,$ppmv with respect to pre-industrial control (averaged over the parameter combinations, similarly to Fig. 3). (b): Bar plot of the anomaly of weathering flux from panel (a) integrated over the geographic regions (similarly to Fig. 3c).

### 3.2.2 Simulation with full Pliocene Sea Surface Temperature

We now examine the slab ocean climate simulations where the full Pliocene SST field is applied. The most noticeable feature

of the full Pliocene SST simulation is its mean temperature anomaly. As shown in Fig. 8a, for any given $CO_2$ level, the full Pliocene SST slab ocean simulation is about $2.5\,°$C warmer than its counterpart with pre-industrial boundary conditions.

These warmer conditions at a given $CO_2$ level can be explained by analysing the components of the radiative budget. Figure 9 shows the comparison between the full Pliocene SST simulation and pre-industrial simulations interpolated at $309.4\,$ppmv (same $CO_2$ level as Pliocene SST simulation) and at $538.3\,$ppmv ($CO_2$ level at which the Global Mean $2\,$m Temperature

equals the one of the Pliocene SST simulation). A significant signal of the full Pliocene SST simulation can be found in the net solar radiative flux at top of atmosphere (Fig. 9 a and e). There are 2 peaks of incoming radiation in the tropics (around $5\,°$N and $10\,°$S, Fig. 9e), that do not appear just by raising $CO_2$ with pre-industrial boundary conditions. Such tropical peaks are caused by reduced cloudiness (Fig. 9 d and h), due to less intense convection because of the reduced zonal temperature gradient, lowering Earth shortwave albedo. Indeed, the cloud contribution to the peaks, estimated by the difference "full sky"

minus "clear sky" (dashed lines in Fig. 9e) is almost 100% of the signal. The SST pattern effect on global cloud radiative feedback has indeed been recognized (e.g., Zhou et al., 2017). Another peak in incoming solar radiation can be seen around $40\,°$ N or S Fig. (9e) and corresponds to a poleward shift in cloudiness at those latitudes (Fig. 9d), mostly discernable in the Southern Hemisphere. This feature, however, is also found in the pre-industrial simulation at $538.3\,$ppmv, and may be a mere consequence of rising global temperature, though its amplitude is higher in the full Pliocene SST simulation. More

solar radiation also enters at high latitude in the Pliocene simulations relative to the pre-industrial control, due to the retreat of sea ice, also lowering shortwave albedo. Though it is is countered by increase in cloudiness (Fig. 9d and h), as can be seen in the cloud contribution in Fig. 9e, the net effect is still positive. Similarly to the peak at $40\,°$ N or S, this phenomenon is

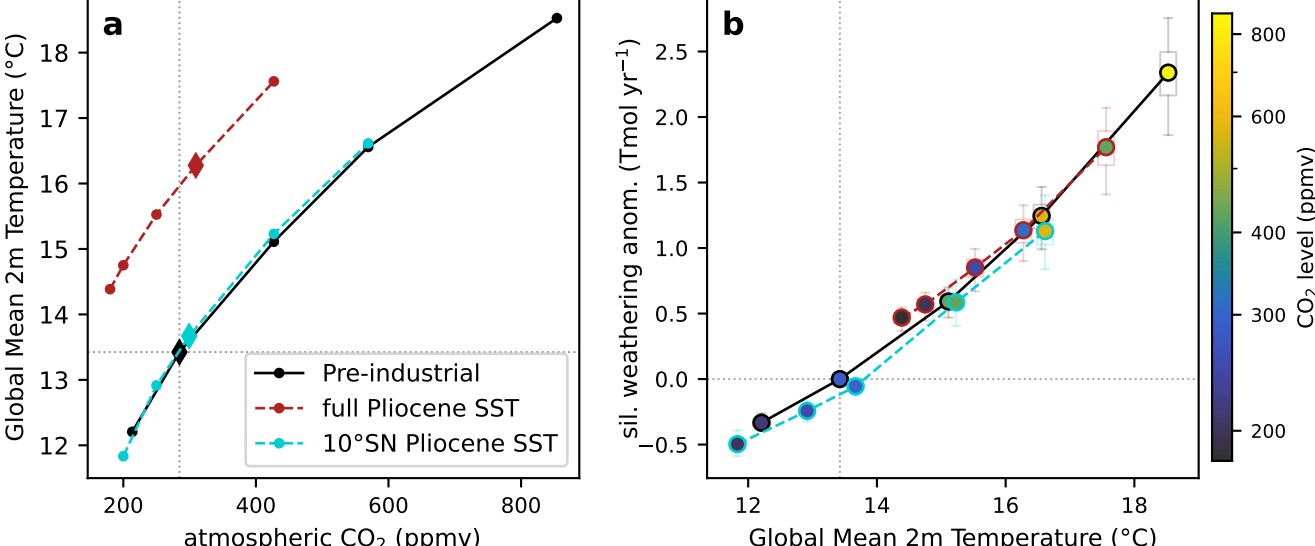

**Figure 8.** (a): Global Mean 2 m Temperature (GMST) plotted against atmospheric $CO_2$ level for the 3 series of slab ocean GCM simulations: pre-industrial boundary conditions, full Pliocene SST and $10\,°$SN Pliocene SST. Each dot represents an actual simulation, the diamonds correspond to the "standard" $CO_2$ levels ($284.7$ ppmv for pre-industrial, see previous section for Pliocene SST). (b): Total silicate weathering fluxes anomaly (with respect to pre-industrial) plotted against GMST, for the same 3 series of slab ocean GCM simulations (same color code than panel a). Each circle represents a single simulation, at fixed $CO_2$, its level being indicated with the colorscale. The semi-transparent boxplots show, for each simulation, the variability of weathering across the parameter combinations. On each panel, null weathering anomaly, pre-industrial GMST and pre-industrial $CO_2$ level are highlighted by the vertical and horizontal thin grey dotted lines.

also present in the pre-industrial $538.3$ ppmv simulation, but strengthened in the Pliocene SST simulation. On global average, the net solar radiation forcing of the full Pliocene simulation (with respect to pre-industrial at same $CO_2$) is $+3.9\,\mathrm{W\,m^{-2}}$, the

tropical part alone being $+1.0$ equivalent $\mathrm{W\,m^{-2}}$ (equivalent if spread on the whole Earth). The latitude profile of downwelling long-wave radiation at Earth surface also shows some variations (Fig. 9 b and f), essentially following the anomaly of water vapour (Fig. 9 c and g). The reduced meridional temperature gradient in the full Pliocene SST simulation is responsible for less water vapour in the tropics, and more water vapour in the extra-tropics, compared to the pre-industrial simulation at same mean temperature (Fig. 9g), though it is mostly visible in the Southern hemisphere. The downwelling long-wave radiation at the

surface exhibits the same behaviour. The more significant drops in the tropics (still compared to the pre-industrial simulation at same mean temperature) is likely due to the reduced Hadley circulation. The averaged downwelling long-wave flux anomaly is not so different between the Pliocene SST simulation and the pre-industrial one at $538.3$ ppmv (respectively, $+14.0$ and $+15.8\,\mathrm{W\,m^{-2}}$), both seem to be a consequence of warmer atmosphere (and amplifier, through a positive feedback loop). In summary, we can conclude that the main warming causes of the full Pliocene SST simulation are the decrease in cloudiness in

the tropics, due to a weaker convection and Hadley circulation, combined with a decrease of cloudiness around $40\,°$ N and S and of sea ice at high latitude, these last two also being part of a positive feedback enhancing the "initial" warming.

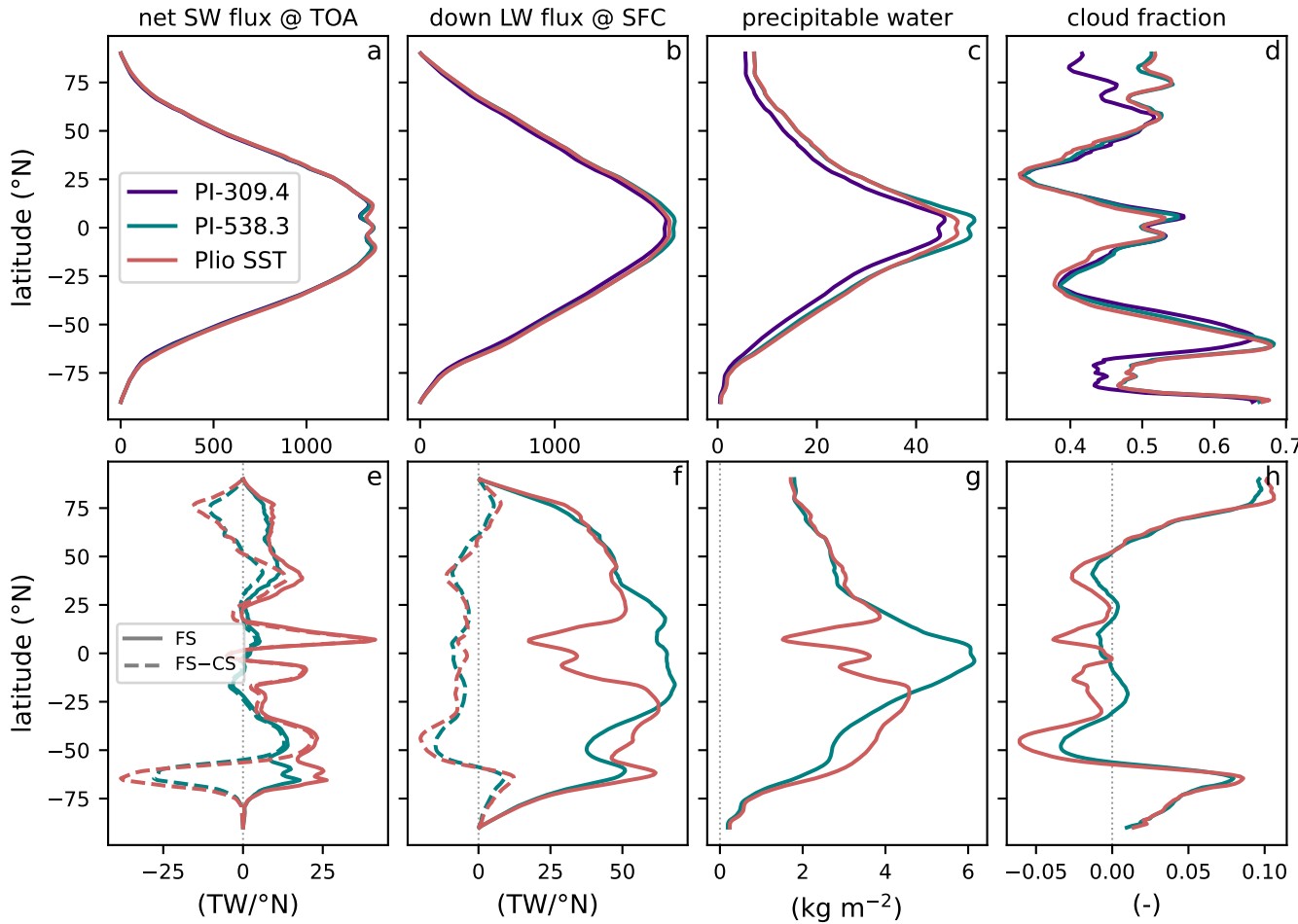

**Figure 9.** Comparison of zonal annual mean fluxes and climatology for the three simulations which are: pre-industrial boundary conditions $\log(CO_2$-interpolated at $309.4\,\text{ppmv}$ (indigo) or at $538.3\,\text{ppmv}$ (teal), and full Pliocene SST (at $309.4\,\text{ppmv}$). (a): zonally-integrated net (incoming minus outgoing) solar (SW) radiative flux at top of atmosphere (TOA), (b): zonally-integrated downwelling long-wave (LW) radiative flux at Earth surface (SFC), (c): zonally-averaged total precipitable water (vertically integrated), (d): zonally-averaged total cloud fraction (vertically integrated). All variables are annual mean climatology. (e), (f), (g) and (h) are similar to (a), (b), (c) and (d) (respectively) but with $309.4\,\text{ppmv}$ pre-indutsrial subtracted, the thin vertical grey dashed line highlights zero anomaly. The dashed color lines in (e) and (f) shows the difference full sky (SF) minus clear sky (CS), illustrating the contribution from clouds. The solid lines (full sky) are the regular values, as in (a) and (b). The "0 anomaly" in panels (e)–(h) is highlighted by a thin grey dotted vertical line. The color code holds for the entire figure.

In term of precipitation, it is easier to compare the full Pliocene SST simulation with the pre-industrial simulation $\log(CO_2)$-interpolated at the same global temperature, to cancel the effect of increased precipitation with global warming. This precipitation (and runoff) difference is shown on Fig. 10. A redistribution of precipitation, with a decrease in the tropics and an increase

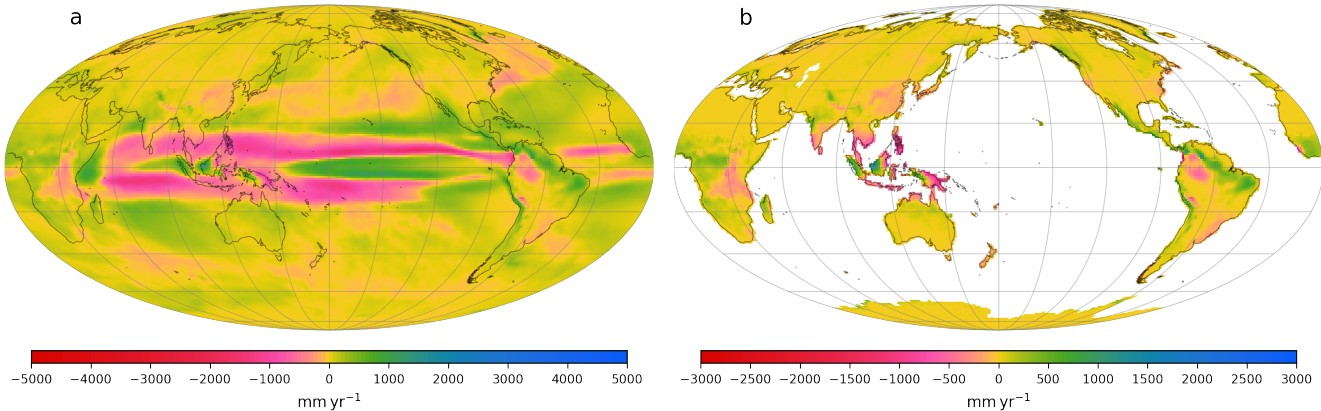

**Figure 10.** Annual mean climatology of the slab ocean simulation with full Pliocene SST at 309.4 ppmv with respect to the pre-industrial control $\log(CO_2)$-interpolated at 358.3 ppmv. This last $CO_2$ level was chosen so that the two simulations in the difference have the same Global Mean 2 m Temperature. (a): total precipitation, and (b): continental runoff. Map projection and graticules as in Fig. 1. On each panel, non-significant differences are hatched (p-value of Welch's t-test > 0.1, see Appendix C)

in the extra-tropics can be observed, superimposed to the anomalies already observed in the $10\,°\mathrm{SN}$ Pliocene SST simulation (Fig. 5): increase in the eastern tropical Pacific and on the western boundaries of tropical Indian and Atlantic oceans. The results are different for land precipitation. Despite the reduced meridional temperature gradient, precipitation and runoff mostly increase in tropical Africa and South America, with the exception of the western part of this Amazon basin and the East African rift, where a drying tendency was already observed in the $10\,°\mathrm{SN}$ Pliocene SST simulation. This goes against our initial hypothesis, based on the "wet gets dryer, dry gets wetter" feature from Burls and Fedorov (2017). The Maritime Continent exhibits contrasting behaviour, with wetter conditions on the western sides of the most equatorial islands, and dryer conditions on their eastern side. Such behaviour was also present in the $10\,°\mathrm{SN}$ Pliocene SST simulation, with a larger precipitation increase on the western sides, but is highlighted here because the anomaly is computed with respect to the pre-industrial simulation at 538.3 ppmv instead of 284.7 ppmv: the increase on the eastern sides is less than the roughly uniform increase due to higher $CO_2$, whereas the increase on the western sides is larger. On global average, precipitation increase by $15\,\mathrm{mm\,yr^{-1}}$ and continental runoff increase by $16\,\mathrm{mm\,yr^{-1}}$.

The offset in global temperature for a given $CO_2$ level (in full Pliocene SST simulations, with respect to pre-industrial ones) poses a technical problem: to go back down to pre-industrial temperature, with full Pliocene SST, $CO_2$ needs to be lowered to $\sim 140\,\mathrm{ppmv}$. Such $CO_2$ levels are unrealistically low and are even lower than those during Pleistocene glacial maxima. Moreover, at this $CO_2$ level, plant stomates cease to operate normally in the land model of CESM, with huge consequences on the hydrologic cycle. Shut stomates behaviour contrasts with the regular tendency for stomates is to be more open at lower $CO_2$ and severely reduces transpiration, eventually leading to increased global runoff, though the mean temperature is lower. Therefore, it is not possible to properly simulate silicate weathering with GEOCLIM. Increased runoff would lead to increased weathering at lower $CO_2$ (thereby killing the negative feedback). This effect is likely responsible of the "flattening" of the

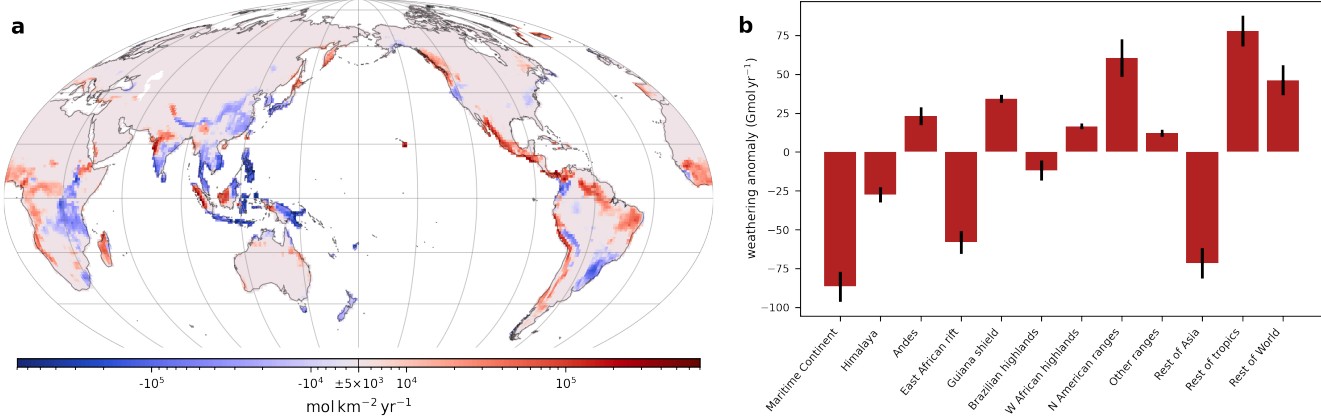

**Figure 11.** (a): Anomaly of weathering rate of slab ocean climate simulation with full Pliocene SST at $309.4\,\mathrm{ppmv}$; anomaly computed with respect to pre-industrial simulation interpolated at $358.3\,\mathrm{ppmv}$ (see Fig. 10); (b): Bar plot of the anomaly of weathering flux from panel (a).

temperature-weathering curve at the lowest tested $CO_2$ levels (dashed red curve on Fig. 8b). On the other hand, the fact that plants cannot survive would undermine weathering, as plant roots are a major weathering agent. One may expect this second effect to be stronger, and hence to see a collapse of total weathering flux with even lower $CO_2$. For this reason, we cannot perform the inversion to compute the equilibrium $CO_2$ where silicate weathering flux balances pre-industrial degassing. We instead analyse the weathering fluxes at fixed $CO_2$.

Figure 8b illustrates the silicate weathering fluxes of the pre-industrial (solid black curve) and full Pliocene SST (dashed red curve) simulations. Despite the higher continental runoff, compared to the pre-industrial simulation at the same global temperature, the full Pliocene SST temperature-weathering curve and the pre-industrial curve are superimposed, except for very low $CO_2$ ($< 200\,\mathrm{ppmv}$) Pliocene simulations. This result means that Pliocene SST does not generate any significant increase or decrease of weatherability – except at low $CO_2$, but likely due to $CO_2$-plants interaction and stomates closure

that generates higher runoff. If the carbon cycle was left to reach balance, with pre-industrial degassing, $CO_2$ would decrease, but it is not possible to determine what would be the equilibrium temperature. The dashed red curves on Fig. 8b cannot be extrapolated on the lower bound because it will falls outside range of validity of the weathering model.

    The fact that higher mean runoff does not lead to higher weatherability means that the regions where runoff decrease matter more in term of weathering. Comparing the full Pliocene SST simulation to the pre-industrial interpolated at the same global

temperature ($538.3\,\mathrm{ppmv}$) reveals weathering decreases in the almost all of the South-East Asia, India, China, the East African rift, and even the Brazilian highlands and the Himalaya (Fig. 11). The combined decrease of those regions – mostly the first four – is enough to counteract the increase in the rest of the tropical land masses (Africa and South America), and along most of the American Cordillera (from the Andes to the Cascade range, Fig. 11a) as shown in Fig. 11b.

## 4   Discussion

Our results emphasize the different effects on silicate weathering of meridional and tropical zonal temperature gradients, that are characteristic of Pliocene warmth, according to temperature proxies and mechanistic connection between meridional and zonal SST gradients (Fedorov et al., 2015).

An estimate of the consequences of Pliocene permanent El Niño using reanalysis of modern El Niño events suggests a reduction of global weatherability by $1.7\%$, that would correspond to a $\sim 0.23\,^{\circ}\mathrm{C}$ warming relative to today (assuming a simple

linear weathering feedback of $\sim 0.4\,\mathrm{Tmol\,yr^{-1}}$ per $^{\circ}\mathrm{C}$ of global warming by $CO_2$ that compensates the $+1.7\%$ weathering anomaly). Climate simulations with a "permanent El Nino" reconstruction of Pliocene SST in the $10\,^{\circ}\mathrm{S}$–$10\,^{\circ}\mathrm{N}$ band indicate a $\sim 0.4\,^{\circ}\mathrm{C}$ warming relative to present-day conditions, but arising from different mechanisms. While El Niño reanalysis emphasized the role of the Maritime Continent in this warming through reduced weathering fluxes, climate simulations suggest in contrast a higher weathering flux in the Maritime Continent, offsetting the warming generated by the weathering decrease in

mainland South-East Asia, India and the East African rift. Determining which of the two scenarios is the most reliable is not an easy task. Climate models contain biases, in particular for simulating precipitation. The way precipitation on the Maritime Continent is affected by SST changes, in the CESM model, may not be accurate. Indeed, the fully-coupled version of the climate model we used simulates rainfall increases in the winter and spring months over the South East Asian Islands following peak El Nino, unlike the weak drying seen in observations (Deser et al. 2012, compare their figures 9 and 10). On the other

hand, modern El Niño consists of transient events evolving over a period of a year and with a peak in winter, and not subject to the constraint of balancing Earth's radiative budget. These features differ from stationary (permanent) El Niño conditions, where the anomalously warm SSTs over the eastern equatorial Pacific occur for all months. Thus, modern-day El Niño events and their teleconnected impacts are likley to be quite different from that of a Pliocene permanent El Niño.

Nonetheless, the results agree on a modest warming effect ($\sim 0.4\,^{\circ}\mathrm{C}$) of permanent El Niño resulting from a reduced

global weatherability relative to the present day. Though its amplitude is small, this warming is robust across all weathering model parameterizations. However, this effect is not seen when the entire estimation of Pliocene SST is applied to the climate model. Further complications arise from the radiative budget of the Pliocene SST simulation that is $\sim 2.5\,^{\circ}\mathrm{C}$ warmer than pre-industrial for a given $CO_2$ level. This temperature shift is mostly caused by lower shortwave albedo due to fewer clouds in the tropics, as well as a poleward shift in the mid-latitude peak of cloudiness, and reduced sea-ice. If a flatter meridional

temperature gradient is the result of warmer global temperature (due to higher $CO_2$), a direct consequence would be that climate sensitivity to $CO_2$ is much higher than current estimates. This raises the question of how representative of Pliocene SST is our reconstruction using Burls and Fedorov (2014)'s method – modifying clouds shortwave albedo – though it is globally in agreement with SST proxies.

The larger lesson we learn from our simulations is that SST pattern changes result in large regional changes in weathering of

both signs. They tend to cancel each other out, but depending on the SST pattern there can still be a significant warming effect, as seen in the permanent El Nino and $10\,^{\circ}\mathrm{S}$–$10\,^{\circ}\mathrm{N}$ Pliocene SST cases. While the effect of reduced meridional SST gradient seems to cancel the reduced tropical zonal SST gradient in our Pliocene SST simulation by increasing mean precipitation and

continental runoff, this cancellation could be seen as fortuitous and that (say) a different magnitude of the reduced meridional SST gradient could give rise to a net effect. A proper evaluation of the weatherability pattern effect for the Pliocene thus requires a robust SST reconstruction for that time period.

## 5   Conclusions

Long-term cooling of Earth's climate from the mid-Miocene to the present associated with decreasing $CO_2$ likely resulted from increased weatherability (e.g., Park et al., 2020), decreased outgassing (e.g., Herbert et al., 2022, though no clear change is noted since the Pliocene), or a combination of the two. Superimposed on this long-term trend are large changes in climate dynamics whose impacts on regional climatology could play an important role in processes such as the growth of Northern Hemisphere ice sheets (e.g., Molnar and Cronin, 2015). Additionally, regional changes in precipitation and runoff could alter silicate weathering with resulting implications for geologic carbon sequestration and steady-state $CO_2$ levels. We coin this phenomenon the "weatherability pattern effect", analogous to the pattern effect in climate sensitivity. In this contribution, we investigated the "weatherability pattern effect" due to SST gradients. We both put forward and evaluate the hypothesis that the transition from a Pliocene permanent El Niño climate to a modern El Niño-Southern Oscillation climate could be associated with an increase in weatherability due to shifts in precipitation to or from chemical weathering hotspots. We explored this hypothesis by running a chemical weathering model with El Niño climate as found in reanalysis data, and by climate model simulations imposing Pliocene-like SST changes. Significant regional changes of both signs are found in the weathering patterns of the different experiments, particularly in tropical weathering hotspots. These changes largely cancel, but can produce a modest but significant weatherability increase should they not entirely compensate. These results highlight the potential importance of the weatherability pattern effect on Earth's long-term carbon cycle, though a proper evaluation requires a robust reconstruction of the relevant SST patterns.

*Code availability.* All code associated with this study is archived on Zenodo (https://doi.org/10.5281/zenodo.8013407); doi: 10.5281/zenodo.8013407

*Data availability.* All climate and GEOCLIM simulations used for this study are available on the dataset https://doi.org/10.6078/D11H7D

## Appendix A:  Calculation of ENSO index in ERA5 reanalysis

We generated the ENSO index from the climate fields from ERA5 reanalysis (1979–2020 time-series) as follows: we used the detrended monthly SST average on Niño 3 region, that is $5\,°N–5\,°S$, $150\,°–90\,°W$ (Trenberth and Stepaniak, 2001). We detrended the time-series by performing a linear regression with respect to time, and subtracted the "time" term from the original time-series. We grouped the monthly time-series in "year" vectors of 12 elements, corresponding to the 12 months,

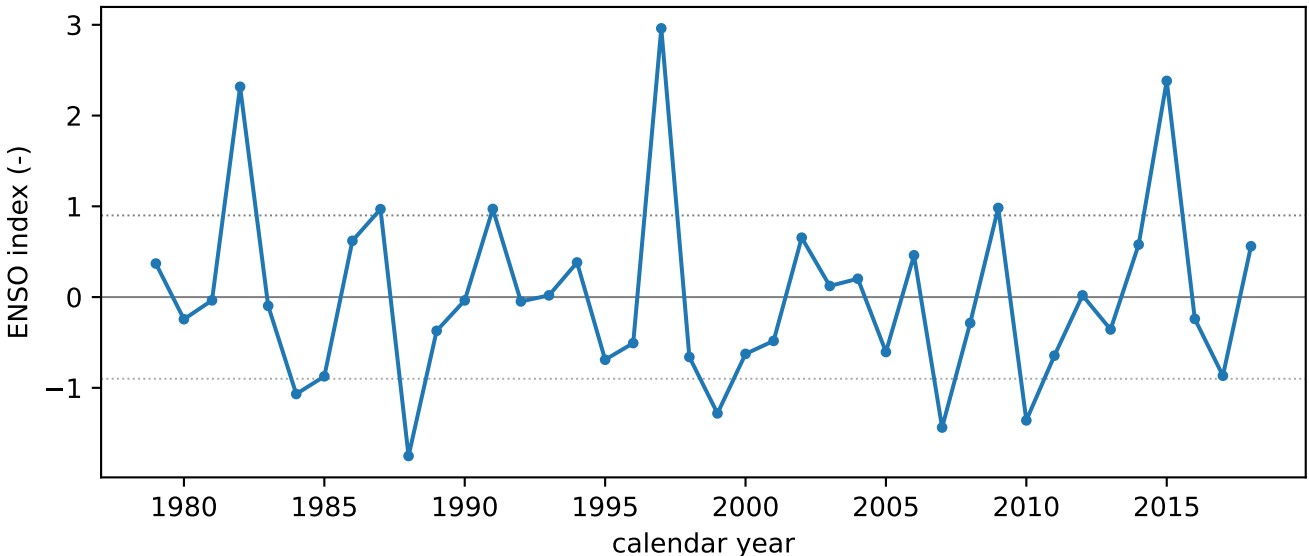

**Figure A1.** Time-series of the calculated ENSO index. The horizontal lines highlight the 0 value and the thresholds values of 0.9 and -0.9 chosen for El Niño and La Niña years respectively.

from May to April of the next calendar year. We subtracted to each year vector the average of all years, to center the vectorial time-series. We then performed a principal component analysis. We identified the first eigenvector (or EOF, for empirical orthogonal function) as the signature of El Niño events. We projected the detrended, centered, vectorial time-series on this first EOF (scalar product of the two 12 elements vectors, for each year). We normalized the scalar time-series obtained in this way
by dividing it by its standard deviation (across the years). This normalized time-series can be interpreted as a time-series of the El Niño index for each year. We selected the El Niño years as the years having an index $> 0.9$, and $< -0.9$ for the La Niña years.

The resulting ENSO index yearly time-series is shown in Fig. A1.

## Appendix B:  Design of slab ocean simulations with Pliocene SST

The aim of the following procedure is to generate slab ocean climate simulations – at several $CO_2$ levels – with SST patterns that replicate the meridional and tropical zonal gradients presented in Burls and Fedorov (2014); Fedorov et al. (2015); Burls and Fedorov (2017). We proceeded in three steps.

### B1   Full ocean simulations at pre-industrial $CO_2$

For this step, we strictly applied the method of Burls and Fedorov (2014) to seek to reproduce Pliocene SST. From this
simulation, we will only consider the SST field.

In the ocean-atmosphere coupled version of CESM1.2, we modified the cloud shortwave albedo by changing the value of the water path in the radiative code of CAM4. The cloud liquid and solid water path multiplied by 3.4 between $15\,°$N North and South, and the cloud liquid water path multiplied by 0.4 for the rest of the Earth. We ran those simulations for 230 years, which is sufficient for the surface ocean to respond to the perturbation (Fedorov et al., 2015). At this point, we are not interested
in accurately reaching the radiative equilibrium. The last 30 years of simulation were used to extract the SST field. This field exhibits the features of reduced zonal and meridional gradients, in accordance with Burls and Fedorov (2014). We refer to this SST field as the "full Pliocene SST". We also ran a pre-industrial control simulation for 170 years, taking the last 30 years to extract the "pre-industrial control" SST field. Finally, in order to create a Pliocene SST field for the tropics only, we merged the full Pliocene SST in the tropics to the pre-industrial control SST in the extratropics using

$$SST_{10SNPliocene} = f SST_{Pliocene} + (1-f) SST_{Preindustrial} \tag{B1}$$

where

$$f = \exp(-(latitude/10)^6) \tag{B2}$$

with latitude in degrees. We refer to this SST field as the "$10\,°$SN Pliocene SST"

**B2  Fixed-SST simulations with zero-integral net ocean-atmosphere heat flux**

In this second step, we took the "pre-industrial control SST", the "full Pliocene SST", and the "$10\,°$SN Pliocene SST" fields, and ran atmosphere-only (i.e., fixed-SST) simulations for each of those fields.

Note that we use the standard CAM4 model for each simulation (i.e. we do not alter the cloud properties), so that the climate model physics is consistent. Those simulations were run at several $CO_2$ levels ($284.7\,$ppmv, $300\,$ppmv and $320\,$ppmv for the full Pliocene SST; $284.7\,$ppmv and $300\,$ppmv for the $10\,°$SN Pliocene SST). We did so in order to estimate at which
$CO_2$ the atmosphere is in equilibrium with the SST, meaning that the net surface heat flux ($F_{net}$, Eq. B3) sums at zero. We estimated those $CO_2$ levels as $309.4\,$ppmv for the full Pliocene SST and $299.4\,$ppmv for the $10\,°$SN Pliocene SST, by linearly interpolation $F_{net}$ as a function of $\log(CO_2)$. $F_{net}$ is computed as:

$$F_{net} = F_{SW} - F_{LW} - F_L - F_S \tag{B3}$$

Where $F_{SW}$ is the net (downward) solar flux, $F_{LW}$ is net (upward) long-wave flux, $F_L$ is the latent heat flux, and $F_S$ the
sensible heat flux, all fluxes at Earth surface. These atmosphere-only simulations were run 40 years, and the last 30 years were used to extract $F_{net}$.

**B3  Slab ocean simulations with Pliocene SST**

In this last step, we used 0-sum $F_{net}$ from the "full" and "$10\,°$SN Pliocene SST" atmosphere-only simulations to derive oceanic Q flux: $Q$. The Q flux refers to the forcing term of a slab ocean model. It represents the divergence of oceanic heat transport,
and directly controls the warming or cooling of the surface ocean, in a slab model.

We computed the Q flux for each month of the annual cycle with the following equation:

$$Q(t) = F_{net}(t) - \rho_{wat}c_{p_{wat}}h_{ml}\frac{d(SST)}{dt} + \rho_{ice}L_{fus}h_{ice}\frac{dx_{ice}}{dt} \tag{B4}$$

Where $\rho_{wat}$ is the seawater density, $c_{p_{wat}}$ is the seawater heat capacity, $h_{ml}$ is the depth of ocean mixed layer, $\rho_{ice}$ is the density of sea ice, $L_{fus}$ is the latent heat of fusion of water, $h_{ice}$ is the thickness of sea ice, and $x_{ice}$ the fraction of each oceanic grid covered by sea ice. The time derivative was approximated here as the month by month finite difference. The mixed layer depth $h_{ml}$ was taken from the default pre-industrial Q flux forcing (see Methods, section 2.3) and is time-invariant. The SST and ice fraction ($x_{ice}$) were taken from the Pliocene SST atmosphere-only simulations (full or $10\,°SN$).

More precisely, we computed the anomaly of $SST$, $x_{ice}$ and $F_{net}$ by subtracting the fields taken from Pliocene SST atmosphere-only simulations to the fields from the pre-industrial control atmosphere-only climate run. Noting this subtraction $\Delta$, we actually computed $\Delta Q$ as:

$$\Delta Q(t) = \Delta F_{net}(t) - \rho_{wat}c_{p_{wat}}h_{ml}\frac{d(\Delta SST)}{dt} + \rho_{ice}L_{fus}h_{ice}\frac{d(\Delta x_{ice})}{dt} \tag{B5}$$

With $\Delta Q$ thus calculated, the Pliocene Q flux was computed as $Q_{control} + \Delta Q$ (for both full SST or $10\,°N/S$ SST fields). Using this generated Q flux, we ran climate simulations with the slab ocean version of CESM1.2, at the "standard" $CO_2$ level (309.4 ppmv or 299.4 ppmv, see previous paragraph) and at higher or lower $CO_2$ (180 ppmv, 200 ppmv, 250 ppmv, 427.1 ppmv and 569.4 ppmv) to encompass the full range of climate warming or cooling, as $CO_2$ must adjust to balance the geological C cycle. Here again, the standard CAM4 model is used in all instances, so that the climate model physics is consistent.

The least constrained variable here is the sea-ice thickness ($h_{ice}$). In the absence of information, as it cannot be retrieved from the atmosphere-only simulations, we assumed a constant, uniform thickness of $1\,m$. Despite this crude assumption, the slab ocean simulations (at standard $CO_2$) reproduce well the SST fields from the original ocean-atmosphere coupled model (see Fig. 4).

We note that these estimated $CO_2$ levels that balance the heat flux in the atmosphere-only simulations are different from the C cycle equilibrium $CO_2$ levels computed by GEOCLIM. The former is done so that the derived Q-flux change does not artificially introduce heat into or out of the ocean. With a slab ocean model, the null $F_{net}$ condition is verified for any $CO_2$ level, because it is imposed by the Q flux. The latter $CO_2$ levels computed by GEOCLIM are the ones that balance the geologic C cycle.

Finally, we ran "full Pliocene SST" and "$10\,°N/S$ Pliocene SST" slab ocean simulations at several $CO_2$ levels using the Q fluxes thus generated. Table B1 summarizes all the climate simulations that were conducted for this study. The slab ocean simulations (last two entries of Table B1) are the main focus of the article, the other simulations are intermediate simulations needed to design the slab ocean simulations.

**Table B1.** Summary of all the climate simulations used for the present study (including "intermediate" simulations) and the specificity of their design.

| Name | boundary conditions* | ocean model | CO$_2$ level (ppmv) | integration time |
|---|---|---|---|---|
| COA-ctrl | pre-industrial | full | 284.7 | 170 yr |
| COA-Plio | modified cloud albedo | full | 284.7 | 230 yr |
| fSST-ctrl | pre-industrial | fixed SST | 284.7 | 40 yr |
| fSST-Plio-full | SST from "COA-Plio" | fixed SST | 284.7; 300; 320 | 40 yr |
| fSST-Plio-10SN | 10 °S–10 °N SST from "COA-Plio" | fixed SST | 284.7; 300 | 40 yr |
| control | pre-industrial | slab | 213.5; 284.7; 427.1<br>569.4; 854.1; 1138.8 | 50 yr |
| full Pliocene SST | Q flux derived from "fSST-Plio-full" | slab | 180; 200; 250<br>309.4; 427.1 | 50 yr |
| 10°SN Pliocene SST | Q flux derived from "fSST-Plio-10SN" | slab | 200; 250; 299.4<br>569.4; 854.1 | 50 yr |

\* refers to all climate model boundary conditions except atmospheric CO$_2$. Anything else than the indicated modifications is pre-industrial

## Appendix C:  Statistical significance tests

Figures 1, 5 and 10 include statistical tests to determine statistically significant patterns in the anomalies. In every case, a Welch's t-test was performed, to compare the "control" and the "perturbed" fields (temperature, precipitation or runoff). For those tests, we considered samples of annual means (i.e., 1-year average), where every year is assumed to be an independent observation. Hence, the variance estimates used for the tests are the empirical variance between the years. The null hypothesis of the test is that the "control" and "perturbed" averages (i.e., average of all the 1-year annual means) are equal.

For the CESM simulations, the "control" population is the 30-year climatology of the pre-industrial simulation, and the "perturbed" populations are the 30-year climatology of either the 10 °SN Pliocene SST (Fig. 5) or full Pliocene SST (Fig. 10) simulations. The sample size $N$ is 30 for both the "control" and "perturbed" populations.

For the ERA5 reanalysis (Fig. 1), the test we performed is meant to compare the selected El Niño years and the "regular" years. Therefore, the "control" population is the sample of all the years (from May to following April) whose ENSO index exceed neither the El Niño nor the La Niña threshold (sample size $N = 29$) and the "perturbed" population is the sample of El Niño years (sample size $N = 6$).

## Appendix D:  Pliocene SST proxies

We used Alkenone SST proxy (Uk'37) from ODP sites 1125 (Fedorov et al., 2015), 850, 806 (Zhang et al., 2014), 1208, 1021 (LaRiviere et al., 2012), 607, 856, 847, 982 and 1241 (Brierley et al., 2009). For each site, we averaged the available SST

**Table D1.** Details of SST proxies shown in the present study

| ODP site | longitude | lattitude | averaging interval | paleo SST | modern SST | reference |
|:--------:|:---------:|:---------:|:------------------:|:---------:|:----------:|:---------:|
| 1125 | $-178\,^{\circ}$E | $-42\,^{\circ}$N | $3.536\,\mathrm{Ma} - 4.485\,\mathrm{Ma}$ | $19.74\,^{\circ}$C | $15\,^{\circ}$C | Fedorov et al. (2015) |
| 850 | $-111\,^{\circ}$E | $1\,^{\circ}$N | $3.73\,\mathrm{Ma} - 4.33\,\mathrm{Ma}$ | $26.97\,^{\circ}$C | $24.9\,^{\circ}$C | Zhang et al. (2014) |
| 806 | $159\,^{\circ}$E | $0\,^{\circ}$N | $3.54\,\mathrm{Ma} - 4.44\,\mathrm{Ma}$ | $28.2\,^{\circ}$C | $28.2\,^{\circ}$C* | Zhang et al. (2014) |
| 1208 | $158\,^{\circ}$E | $37\,^{\circ}$N | $3.52\,\mathrm{Ma} - 4.49\,\mathrm{Ma}$ | $21.72\,^{\circ}$C | $17.5\,^{\circ}$C | LaRiviere et al. (2012) |
| 1021 | $-128\,^{\circ}$E | $39\,^{\circ}$N | $3.53\,\mathrm{Ma} - 4.48\,\mathrm{Ma}$ | $15.01\,^{\circ}$C | $11\,^{\circ}$C* | LaRiviere et al. (2012) |
| 607 | $-33\,^{\circ}$E | $41\,^{\circ}$N | $3.9\,\mathrm{Ma} - 4.1\,\mathrm{Ma}$ | $21.5\,^{\circ}$C | $18.5\,^{\circ}$C | Brierley et al. (2009) |
| 846 | $-91\,^{\circ}$E | $-3\,^{\circ}$N | $4.0\,\mathrm{Ma} - 4.3\,\mathrm{Ma}$ | $27\,^{\circ}$C | $23.5\,^{\circ}$C | Brierley et al. (2009) |
| 847 | $-95\,^{\circ}$E | $0\,^{\circ}$N | $4.0\,\mathrm{Ma} - 4.3\,\mathrm{Ma}$ | $28\,^{\circ}$C | $24.5\,^{\circ}$C | Brierley et al. (2009) |
| 982 | $-16\,^{\circ}$E | $58\,^{\circ}$N | $3.9\,\mathrm{Ma} - 4.1\,\mathrm{Ma}$ | $17.5\,^{\circ}$C | $11\,^{\circ}$C | Brierley et al. (2009) |
| 1241 | $-88\,^{\circ}$E | $3\,^{\circ}$N | $4.0\,\mathrm{Ma} - 4.3\,\mathrm{Ma}$ | $28.5\,^{\circ}$C | $24.5\,^{\circ}$C | Brierley et al. (2009) |

\* taken from the SST proxy time-series (see text)

estimates between $4.5\,\mathrm{Ma}$ and $3.5\,\mathrm{Ma}$ as "Pliocene" SST. In the case of Brierley et al. (2009) (sites 607, 856, 847, 982 and 1241), we took the SST already averaged in Table S1 of their contribution (local SST without correction). For the "control" SST values, we used the modern SST from Fedorov et al. (2015), Table S1 of their contribution. There are two exceptions: for sites 1021 and 806, we considered the near-modern interglacial SST of the time-series as modern SST, because they significantly differ from the actual modern SST (Fedorov et al., 2015, Table S1).

*Author contributions.* P.M. and J.C.H.C. conceived the experiments and ran the climate simulations, P.M. ran the GEOCLIM simulations, wrote the original manuscript draft, and drafted the figures. J.C.H.C. and N.L.S.-H. contributed to manuscript writing. J.C.H.C and N.L.S.-H. acquired funding and supervised project research.

*Competing interests.* The authors declare no conflict of interest

*Acknowledgements.* Research was supported by NSF Frontier Research in Earth Science grant EAR-1925990 awarded to N.L.S.-H. and J.C.H.C. We thank Natalie Burls for useful discussions on climate dynamics and guidance on the design and interpretation of the climate simulations. We dedicate this work to the memory of our dear colleague and friend Sarah White, whose work on permanent El Niño and its impacts inspired this manuscript.

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
