# Peer review of "The effect of the Pliocene temperature pattern on silicate weathering and Pliocene-Pleistocene cooling"

_EGUsphere, 2023_

## Author Comment (AC1)

This study investigates the silicate weathering and carbon cycle for the Pliocene climate that featured a reduced SST gradient in both zonal and meridional directions (the so- called "permanent El Niño"). The authors first explore the impact of present-day El Nino events on the weathering flux using the present-day observation/analysis. Their analysis suggests a small net impact (~2–3%) of ENSO SST patterns on the weathering flux. Next, using climate model simulations, the authors find a similarly small net impact of Pliocene- like SST patterns on the weathering flux.

The research question of the manuscript is very relevant for understanding the global cooling since the Miocene and Pliocene. The introduction is very well written, reflecting that the authors have very good understanding of the problem and relevant dynamical processes. However, I must admit that, to understand the manuscript (such as the motivation and the complicated procedure for creating the Pliocene slab ocean simulation with altered ocean q-flux), I had to read the manuscript several times. This makes me wonder whether the authors can work out a simpler approach to highlight the key experiments and results. Please see also below for a few other comments. These comments need to be addressed before the publication of the manuscript.

Major comments

- Title. Since the main conclusion of the manuscript seem to be that the weathering flux change from the Pliocene-like SST pattern is small in global mean, I think the authors could use a better title to clearly deliver the message, such as "A potentially small net effect of the Pliocene temperature pattern on the silicate weathering and the Pliocene- Pleistocene cooling".

We have modified the title to now be "The effect of the Pliocene temperature pattern on silicate weathering and Pliocene-Pleistocene cooling".

The main advance of our study is in introducing and exploring the weatherability pattern effect to global silicate weathering changes. As our El Nino and reduced zonal gradient examples show, the magnitude of regional silicate weathering changes resulting from SST pattern changes are sizable. If they do not cancel, there is a significant net imbalance to the carbon flux resulting the Earth system to reach a new carbon and temperature equilibrium.

The small net effect seen in our Pliocene simulation should not be interpreted as what happens in the Pliocene – there are too many uncertainties, from the SST pattern reconstruction to climate model simulation of rainfall to the GEOCLIM model. In this sense, the cancellation in the full Pliocene simulation can be seen as fortuitous, and a proper evaluation of the weatherability pattern effect requires (as pointed out by reviewer 3) a robust reconstruction of Pliocene SST patterns.

- Related to 1, the authors conclude that "We find support for this hypothesis…" (Line 416). Given the small net contribution from the SST pattern effect and the abundant uncertainties from SST pattern reconstruction, precipitation bias from model/reanalysis, assumptions built in the silicate weathering model, I wouldn't conclude that "We find support for this hypothesis". The authors' analysis actually lends little support for the hypothesis.

We rewrote a bit the conclusion, and state instead (line 411–417): "We explored this hypothesis by running […] Significant regional changes of both signs are found […] These changes largely cancel, but can produce a modest but significant weatherability increase should they not entirely compensate. These results highlight the potential importance of the weatherability pattern

effect on Earth's long-term carbon cycle, though a proper evaluation requires a robust reconstruction of the relevant SST patterns."

- The manuscript is too long with too many figures (17 figures!), which may decrease the readability and reduce the focus of the manuscript. One way to improve, I think, is to reduce the number of main figures though combining multiple figures into a single figure with a common theme. For example, Figures 2–4 could be combined into one figure showing the average chemical weathering rates and the separation into different regions. Similarly, Figures 5 and 6 could be combined with a focus on the visual and quantitative comparison of the weather rates between El Nino and La Nina events. Same for Figures 10 and 11, and for Figures 16 and 17.

This is a useful advice. In the revised article, we merged figures 2–4 in new Fig. 2, figures 5 and 6 in new Fig. 3, figures 10 and 11 in new Fig. 7, figures 12 and 15 in new Fig. 8, and figures 16 and 17 in new Fig. 11. The revised manuscript now has 11 figures, plus 1 in Appendix A.

- Table 1 and the discussion on the Pliocene experiments are lengthy and difficult to follow. I am sure the authors have carefully thought about the purpose of these experiments, but it is not necessary to list and show all the experiments. For example, the "control" simulations with CO2 of 1138.8 and 854.1 ppmv are not discussed in the manuscript (854.1-ppmv simulation is shown in Figure 12 but may not be needed); same is true for a few other simulations. I think it will help readers to follow and help highlight the most important findings of the manuscript if the authors could carefully examine their simulations and delete unnecessary ones.

We agree that former Table 1 and section 3.2 were difficult to follow, and moving the reader away from the main focus of the manuscript (this point was also raised by reviewer #2).
We restructured the manuscript to keep the focus the 3 series of slab ocean simulations (control, Full Pliocene SST and 10°SN Pliocene SST). We now only briefly present the coupled simulation, and mention the fixed SST as a necessary intermediate step. All the details were moved to Appendix B. Therefore, we did not judge it necessary to keep Table 1 in the main text, and moved it to Appendix B (now Table B1).

- The authors' simulations highly rely on the SST pattern adapted from Burls and Fedorov (2014), but they failed to provide information on how well the SST pattern match the proxy record. I suggest the authors adding values of the meridional and zonal SST gradients from the simulations and the comparison with proxy data. Values of the SST gradients should be provided for COA-ctrl, COA-Plio and all the slab ocean simulations, with Pliocene values compared with proxy reconstruction. This information could be added to Table 1. Also, the global mean surface temperature of each simulation could also be added to Table 1. These key features will help readers better understand each simulation. A short discussion on the implication/caveats should be added if the simulated SST pattern does not match the proxy reconstruction.

We added the alkenone SST proxies presented in Fedorov et al. (2015). Appendix D indicates the source of the data, the averaging interval we considered for Pliocene, and the values (in Table D1). We added the proxy SST anomalies (with respect to modern) to the map in Fig. 4 (former Fig. 7). The coupled ocean-atmosphere simulation does reproduce the high latitude and East Pacific warming of the proxies (which we now indicate lines 229–232) although the East Pacific only warms up by 2.5°C instead of ~4°C in the proxies (see also extra figures at the end of the rebuttal).

Since we moved Table 1 to the Appendix B, we chose not to add information, to keep it a mere description of how the simulations were design. We believe that the SST map and proxies (Fig. 4 a and b) provide enough information regarding the SST gradients.

Finally, we point out that the main goal of our study is in introducing the weatherability pattern effect to global silicate weathering changes, using SST patterns motivated from the Pliocene to explore this hypothesis.  In this sense, our study should be seen as a 'proof of concept' and that a proper evaluation of the Pliocene case requires a robust reconstruction of said SST pattern changes.

- Please add significance test for many of the analysis, such as the difference map in Figures 1, 5, 8, 10, 14, and 16.

Performing statistical tests is possible for some of the difference maps, but not all them.

For the GCM simulations (former Figs. 8 and 14, now Figs. 5 and 10), there is no methodological issue. Each 30-years climatology output can be considered as a N=30 sample of independent observations (i.e., 30 one-year averages). The question "is the average of the 'perturbed' sample statistically different from the average of the 'control' sample" is then relevant, and can be answered with a Welch's t-test (which is what we did).
This approach consider that the relevant entity is the annual mean. Indeed, it does not make sense to determine whether the anomaly caused by the permanent flattening of meridional and zonal gradient can be achieved through the natural intra-annual, or day-to-day variability of climate.

For the ERA5 reanalysis, the question is not as straightforward, because there are not 2 distinct equal size samples. El Niño is a part of the natural variability of climate. So if the question is "Can an El Niño anomaly be achieved by 'chance'", the answer is yes. One cannot reject the null hypothesis because El Niño belongs to the natural variability.
We needed to formulate the question differently: "can the El Niño anomaly be achieved through the natural variability of climate not considering the ENSO variability". In other words, we sought to determine if the average of El Niño years is statistically different from the average of non El Niño nor La Niña years. We designed the statistical test this way. We modified Fig. 1 to show the difference between "El Niño years average" and "non El Niño nor La Niña years average" (instead of "El Niño years" minus the whole time-series average, in the former manuscript), and we performed a Welch's t-test between the N=29 sample of non El Niño nor La Niña years and the N=6 sample of  El Niño years.

The design of these statistical tests is explained in in Appendix C of the revised manuscript.

For the weathering results (former Figs. 5, 10 and 16, now Figs. 3, 7a and 11a), it is not possible to design a statistical test.
There are various sources of uncertainties (or variability) that should be considered. The one we show with error bars on (new) Figs. 2C, 3c, 7b and 11b, or boxplots on Figs. 6 and 8b is the uncertainty associated with the parameters of the weathering model. We considered 573 selected parameter combinations that fitted the data with $r^2 > 0.5$ (cf Park et al., 2020), however, this ensemble can not be considered as a sample of 573 independent and identically distributed observations.
Aside from the parameters uncertainty, there is also the uncertainty due to the natural variability in the climate simulations (or reanalysis) whose outputs are fed to the weathering model. Because of the non-linearity of the model, it is not possible to analytically propagates the variance of the temperature and runoff fields in to the weathering field. This variability will also interact non-linearly with the uncertainty associated with the weathering parameters.

For these reasons, we are not able to provide statistical significance test for the weathering model outputs.

- At many places, when summarizing the effect of weathering flux changes, the authors used the estimated temperature changes, such as Lines 10–11. The authors need to clarify how the temperature changes are estimated and what is the associated uncertainty.

We provided such summarized weathering effect as a temperature change in the abstract (lines 10–11), at the beginning of section 3.1.2 (formerly line 169, now lines 177–178), in section 3.2.1 (formerly line 283, now line 263), and at the beginning of the discussion (section 4, formerly lines 381–382, now lines 369–372). Section 3.2.1 was indeed the only place where we describe how this temperature change is computed (inversion that GEOCLIM performs to find the equilibrium CO2 level at which silicate weathering balances the imposed CO2 degassing), while the previous statement (section 3.1.2) gave little explanation. We added (lines 173–178) a couple of sentences explaining how we converted the weathering anomaly computed with ERA5 reanalysis into a global temperature change, and what is the assumption behind that conversion (linear weathering feedback, with an increase of global weathering flux by 0.4 Tmol/yr per °C of global warming by CO2, compensating the "initial" weathering anomaly).
In the discussion, we remind this assumption of linear weathering feedback (lines 369–371)
In section 3.2.1, the inversion perform by the model to compute the equilibrium CO2 was already explained (lines 275–280 of former manuscript), so we did not modify the text (now lines 255–260).

- When explaining the SST offset between Pliocene and preindustrial slab ocean simulations (Lines 305–351), the authors could mention the SST pattern effect on the cloud radiative effects (such as Zhou et al., 2017, doi:10.1002/2017MS001096).

This is a good suggestion. We added this reference lines 300–301.

Minor comments

- Line 13: Spell the "C cycle" fully.

We spelled "carbon cycle" fully there (now line 14), and a few lines above (line 8)

- Line 27: Change "net net (upward)" to "net (upward)".

Done (former line **227**, now line 464, in Appendix B2)

- Line 205: Change "seeks" to "seek"

Done (now line 439, in Appendix B1)

- Line 310: Change "than" to "as"

Done (now line 294)

- Line 320: Change "visible" to "shortwave"

We have changed "visible albedo" to shortwave albedo in all the occurrences (line 199, line 306, line 388, line 392 and line 441).

[Figure]

Map of SST anomaly (with respect to pre-industrial) of the coupled ocean-atmosphere simulation with altered clouds visible albedo, and Pliocene SST proxy. This figure is identical to main text Fig. 4A, with the name of the ODP sites added.

[Figure]

Anomaly (with respect to pre-industrial) of zonally-averaged SST of the coupled ocean-atmosphere simulation with altered clouds visible albedo, and SST proxy.

[Figure]

Anomaly (with respect to pre-industrial) of SST meridionally averaged between 5°S and 5°N, in the coupled ocean-atmosphere simulation with altered clouds visible albedo, and SST proxy (5°S-5°N ODP sites).

---

## Author Comment (AC2)

This paper uses historical climate reanalysis, paleoclimate simulations and a geochemical weathering model to explore the effect of ENSO on silicate weathering, aiming to test the idea that Pliocene permanent El Nino conditions led to changes in Earth's surface weatherability, such that removal of these conditions may have contributed to long-term CO2 removal and cooling. This is an interesting idea, and the relatively minor effect that is found seems reasonable.

I think the paper can be published with some minor revision. My main points are:

- While the text starts brilliantly, the discussion and conclusions need much more proofreading. I found a lot of gramattical issues in this seciton.

We carefully proofread these sections. A significant part of these sections was also rewritten.

- Section 3.2 which introduces the climate modelling strategy should be explained more clearly. It is really tough to get through at the moment. I would recommend a paragraph at the start of this section clearly setting out what they ultimately want to simulate and how they will do it, before going into the details.

We rewrote section 3.2 to keep the focus on the slab ocean simulations, that are what we ultimately want to simulate. Most of the former section is now in Appendix B. We hope that the text gained in readability.

Minor comments:

- Given that this is a more interdisciplinary paper which focuses mostly on chemical weathering, the authors might consider a higher level explanation of the walker circulation, Bjerknes feedback and southern oscillation.

We added, in the introduction, a sentence shortly explaining what is the Walker circulation (lines 30–32), and another for the Bjerknes feedback (lines 33–34), at the first mention of both.

- Biotic weathering enhancement may play a role here? Plants are very dependent on water. Could the authors acknowledge this? Does their modelling hint at how strong this effect might be?

While plants are not explicitly represented in the weathering model, their effect on weathering (at least a part of it) is likely implicitly incorporated in its parameters, because of the parameter optimization to fit the data. For instance, the parameter $k_w$ describes the sensitivity of weathering rate to runoff rate, and the parameter $k_d$ is also linked to the sensitivity of weathering rate to climatic conditions. The model "learned" the sensitivity of weathering rate to runoff through measurements of weathering in natural systems (rivers). Those systems largely exhibit higher weathering rates with higher runoff, and part of it is because of biotic weathering enhancement (in addition to reactive-transport considerations). Therefore, this process, present in the data, was integrated in the parameters that are optimized to fit the data.
What the weathering model does not represent is the direct effect on weathering of plant fertilization by CO2 (like increased GPP and production of organic acids), for the simple reason that all the data were collected at current (late 20[th]/early 21[th] century) atmospheric CO2. We, however, want to stress that the

effects of plant fertilization by CO2 on evapotranspiration, and hence runoff, is represented by the land module (CLM) of the CESM climate model, so at least a part of the overall "CO2 fertilization effect" is implicitly integrated in GEOCLIM.
Nonetheless, because of this implicit integration rather than an explicit representation, there is no way to determine how strong the effect of biotic weathering enhancement is.

We briefly acknowledged this implicit consideration of vegetation in section 2.1, describing the weathering model (lines 112–115).

- Line 205: "we apply the method of Burls and Fedorov (2014) to seeks to reproduce Pliocene SST." A typo here?

Done (now line 439, in Appendix B1)

- First part of section 3.2 and figure 7 could be explained more clearly. It is not clear at the start of this section what the motivation is for producing these different sets of runs with the full or tropical-only SSTs, and what the comparison in figure 7 is really showing. Some introductory text would be useful.

Section 3.2 was rewritten, it now solely focus on the coupled ocean-atmosphere simulation and the slab ocean simulations (the intermediate simulations needed to design the slab ocean ones are presented in Appendix B). Similarly Fig. 4 (former Fig. 7) now shows the SST from the coupled ocean-atmosphere simulation and the SST-like field (skin temperature) of the slab ocean simulations. We hope that the two aims of Fig. 4: comparing the coupled ocean-atmosphere SST to the SST proxy, and assessing how well the slab simulations reproduce the coupled simulations SST, are more evident in the text (lines 227–237) and in Fig. 4 caption.

- Line 367 "If carbon cycle was left to reach balance", missing word? E.g. "the carbon cycle"And "CO2 would decrease until temperature is back around pre-industrial one" perhaps change to "CO2 would decrease until temperature returned to the pre-industrial value"

This is a good suggestion. This sentence was however modified following comments of Reviewer #3. It is now "If **the** carbon cycle was left to reach balance, with pre-industrial degassing, CO2 would decrease, but it is not possible to determine what would be the equilibrium temperature." (lines 355–356).

- Line 377: "meridional and tropical zonal gradient of temperature" change to "meridional and tropical zonal temperature gradients"?

Done (now line 365)

- Line 394: "the results agree on a moderate warming effect of permanent El Niño". It would be useful to re-state the actual temperature increase here. Also, this is described as minor in the abstract and I would agree. Moderate, in the context of Pliocene warmth, would probably be more like 1C?

We replaced "moderate" by "modest" and state the value of the warming (0.4°C) (now line 384). We also added the sentence "Though its amplitude is small, this warming is robust across all weathering model parameterizations" (lines 385–386).

- Line 396: "that is ~ 2.5 C than pre-industrial", missing word "warmer"?

Indeed. This was corrected (now line 387)

- Line 398: "If flatter meridional temperature gradient", missing word "the"?

We replaced the sentence by "if **a** flatter meridional temperature gradient" (now line 389)

- Line 400: "This raise the question", typo "raises"

Corrected (now line 391)

- Line 418: "the difference of a permanent El Niño climate state on global silicate weathering". Suggest change "on" for "for"

This sentence was removed when rewriting the conclusion.

- Line 421: "particularly on tropical weathering hotspots", change "on" to "in"

Done (now line 414)

---

## Author Comment (AC3)

The authors proposed an interesting hypothesis that the SST pattern in the warm Pliocene has contributed to weaker weatherability and thus helped to maintain the warmer Pliocene climate. To test their hypothesis, the study first calculated weathering fluxes in El Nino years versus La Nina years using reanalysis data. They find reduced (silicate) weathering fluxes during El Nino years due (mainly) to a shift in the precipitation pattern. Next, a set of "Pliocene" simulations were created to further explore how SST patterns in the Pliocene might have affected the weatherability. The results show that regional increases and decreases in weathering fluxes largely cancel out one another. Overall, I am not fully convinced that the Pliocene simulations used in this study well reproduced the Pliocene climate. However, the study highlighted the importance of the SST pattern and mean climatic state on silicate weathering. A valuable lesson is that robust global SST reconstructions integrated with climate simulations are critical in evaluating Earth's thermostat. I propose minor revisions before acceptance.

Major comments:

- Line 213-264: "In order to create a Pliocene SST field for the tropics only". It seems to me this is a scenario with a zonal SST gradient similar to the Pliocene while maintaining a modern meridional SST gradient. But what is the motivation and justification to create an SST field for the tropics only? And why results from this tropical-SST scenario were not included in Figure 15?

This experiment is indeed a way to generate a zonal SST gradient similar to the Pliocene while maintaining a meridional SST gradient. The motivation for isolating the "close-to-equator" zonal SST gradient is that El Niño events consist in the collapse of the tropical Pacific zonal gradient of SST, without major changes in extra-tropical SST patterns while Pliocene climate proxies indicate a reduction of both gradients. Yet, El Niño is viewed as analogy for Pliocene climate, in particular, the same teleconnections are assumed to occur (Molnar & Cane 2002). Isolating the "close-to-equator" SST pattern allows us to investigate the teleconnections caused to the tropical part of the Pliocene SST estimate, without the effects of the reduced meridional gradient, to better compare it to El Niño events. We added this justification in section 3.2 of the revised manuscript (lines 219–226)
We also merged former Figs. 15 and 12 into Fig. 8, in the revised manuscript. This new figure shows the CO2-temperature and the temperature-weathering relationships for both of the Pliocene SST simulations (and the pre-industrial simulation).

Molnar, P. and Cane, M.A., 2002. El Niño's tropical climate and teleconnections as a blueprint for pre-Ice Age climates. Paleoceanography, 17(2), pp.11-1.

- Line 352-362:
  "to go back down to pre-industrial temperature, with full Pliocene SST, CO2 needs to be lowered to ∼140 ppmv."
  "For this reason, we cannot perform the inversion to compute the equilibrium CO2 where silicate weathering flux balances pre-industrial degassing. We instead analyse the weathering fluxes at fixed CO2."

  This paragraph raises some puzzles. Figure 15 seems to suggest that both global temperature and pCO2 will be much lower in the Pliocene in order to balance the pre- industrial CO2 degassing rates. The inference is that (silicate) chemical weathering fluxes must be higher in the Pliocene given the fact that Pliocene was warmer. However, there is no evidence suggesting that Pliocene CO2 degassing rates were higher than today. Based on Figure 15, if we assume Pliocene has the same pCO2 as today (correspondingly ~2.5C warming), the silicate weathering

anomaly is almost ~1 T mol/yr. This is almost a 20% increase from the modern value (Line 153), a very large number for the long-term carbon cycle. If we extrapolate the red curve (full-Pliocene-SST scenario) to the level where weathering flux anomaly = 0, the global temperature probably will be (much) lower.

Thus, the full-Pliocene-SST scenario seems to produce a combination of climate-and-weatherability that is inconsistent with geological evidence. This leads me to suspect that either the simulated Pliocene climate is a poor representation of the Pliocene climate or the sensitivity of GEOCLIM to climate changes needs some revision.

Furthermore, if we extrapolate the red curve to modern global temperature, it seems to suggest a positive anomaly in weathering fluxes. This would indicate a higher weatherability in the Pliocene, as opposed to what Molnar and Cronin proposed? Am I missing something?

My sense is that discussions in lines 380-405 are trying to address some of these puzzles. I would recommend the authors reorganize the discussions in sec 4 a little bit and make a tighter link to Figure 15.

The points raised by the reviewer are true, to a certain extent.

One important element of answer is that we cannot extrapolate the red curve of Fig. 15 (now Fig. 8b) to the level where the weathering flux anomaly = 0. This was discussed in the former manuscript lines 355–358 (ending by: "Therefore, it is not possible to properly simulate silicate weathering with GEOCLIM"). We now explicitly state that the curves cannot be extrapolated (lines 356–357). This extrapolation makes no sense because CO2 would be too low, plants behaviour will radically change: shutting stomates will mean a drastically reduced GPP, so weathering would behave differently than the model is able to computes. One should expect weathering rates to drop because weathering reaction are fostered by plants. So this would mean a lower weatherability, the "red curve" would drop below the black curve, and the temperature where the weathering flux anomaly = 0 may actually be higher than pre-industrial, depending on where exactly this drop would occur, and how strong it would be. However, there is currently no way to quantitatively predict those values. Furthermore, determining them is not necessary relevant, knowing that there are strong evidences that such low CO2 level was never reached in the recent past.

The question "does the full Pliocene SST simulation have a higher weatherability than the pre-industrial" therefore does not have a clear answer. Around 15°C of global temperature, it seems it does (the red curve is above the black curve), but at higher temperature (> 16°C), the 2 curves are superimposed, so the weatherability is the same (as it was already stated in the former manuscript, lines 364–366). And at lower temperature (and CO2), the weatherability should theoretically be lower the pre-industrial for the reason explained here-above.

Concerning the fact that if we assume that CO2 is the same than pre-industrial, weathering flux would be 20% higher; it is true indeed, as temperature is 2.5°C higher and the weatherability is unchanged (in our simulations). This simply means that the "full Pliocene" SST pattern cannot be – carbon cycle mass balance in mind – the reason why Pliocene was 2.5°C warmer, and there must be another reason. Such other reasons could be a reduction of weatherability due to less emerged area in the Maritime continent (e.g., Molnar & Cronin 2015, Park et al., 2020), or another C flux that was reduced (for instance less organic C burial)…

Section 3.2.2 was partially re-written to make those points more evident (lines 343–357).

- Line 210: Can the authors comment on how well the simulated SST agrees with proxy data, rather than simply citing Burls and Fedorov, 2014? For instance, what is the magnitude of warming in the EEP relative to the proxies?

We added SST proxies from 10 ODP sites (following Burls and Fedorov, 2015). The Pliocene SST anomaly from these proxies is now shown on Fig. 4 (former Fig. 7). The agreement with the proxy is also discussed lines 227–232 The details on the ODP sites, source studies, and averaging time interval is given in Appendix D, and Table D1.

The warming of the Tropical East Pacific is the coupled ocean simulations is about half of what is observed in the proxies (~2.5°C versus ~4°C, see also extra figures at the end of the rebuttal).
This remark made us notice that we already provided a value: 2°C, for the Tropical East Pacific warming (actually, the reduction of the west-to-east Pacific gradient), line 284 of the former manuscript. This value is closer to 2.5°C, so we corrected that sentence (now line 265)

Some minor points:

- The authors may want to mention at the beginning that there is little evidence of changes in CO2 degassing rates since the Pliocene. Thus, a change in weatherability (e.g. Molnar and Cronin) is the most likely explanation for the long-term coolling.

This is a good suggestion. We added this information lines 23–25. We also modify the conclusion, when we mention decreasing degassing as a potential driver (formerly line 408, now lines 403–404), to specify that "no clear change is noted since the Pliocene".

- Line 68-69: It seems to be a little bit controversial now to say "El Niño events is a good representation of Pliocene permanent El Niño."

This is an accurate remark, given our conclusions. We replaced it by "with the assumption that the average climate of El Niño events can be used as a proxy for Pliocene permanent El Niño" (now line 74–75).

- Line 71-73: "However, one cannot quantitatively estimate this warming … yet, if silicate weathering is disturbed, CO2 will adjust …" This sentence can be confusing to some readers. Maybe reword it a little bit?

We split the sentence in two, and added more precision (now lines 77–80)

- Line 106-122: consider to provide a supplementary figure showing the time series of the ENSO index you calculated?

We moved the details of the calculation of the ENSO index (former lines 110–120) from section 2.2 to Appendix A, and added a figure (Fig. A1) showing the index time-series.

- Line 155: probably point out what climate field they used.

We added this information (now lines 158–159)

- Line 366: "This result means that Pliocene SST does not generate any significant increase or decrease of weatherability, save perhaps at low CO2" typo?

This sentence was changed (as the section was partially re-written) to "This result means that Pliocene SST does not generate any significant increase or decrease of weatherability – except at low CO2 [...]" (lines 353–354).

- Line 410: "could plan an important role" typo?

Indeed. This was corrected (now line 405)

- Line 345: "This goes against our initial hypothesis, based on the "wet gets dryer, dry gets wetter" feature from Burls and Fedorov (2017)." I think this is the first time you mention this. It is better to front load this in the introduction.

We meant to refer to the behaviour of reduced equatorward moisture transport (lines 49–50 of the former manuscript, now lines 54–55), that is at the root of the "wet gets dryer, dry gets wetter" feature. In the revised manuscript, we explicitly mention that feature in the introduction, after mentioning the reduction of equatorward moisture transport (lines 55–56).

[Figure]

Map of SST anomaly (with respect to pre-industrial) of the coupled ocean-atmosphere simulation with altered clouds visible albedo, and Pliocene SST proxy. This figure is identical to main text Fig. 4A, with the name of the ODP sites added.

[Figure]

Anomaly (with respect to pre-industrial) of zonally-averaged SST of the coupled ocean-atmosphere simulation with altered clouds visible albedo, and SST proxy.

[Figure]

Anomaly (with respect to pre-industrial) of SST meridionally averaged between 5°S and 5°N, in the coupled ocean-atmosphere simulation with altered clouds visible albedo, and SST proxy (5°S-5°N ODP sites).